# A vegetation phenology dataset by integrating multiple sources using the Reliability Ensemble Averaging method

Yishuo Cui [1], Shouzhi Chen[1], Yufeng Gong[1], Mingwei Li[1], Zitong Jia[1], Yuyu Zhou[2], Yongshuo H. Fu[1,3]

[1]College of Water Sciences, Beijing Normal University, Beijing 100875, China
[2]Department of Geography, The University of Hong Kong, Hong Kong, China
[3]Plants and Ecosystems, Department of Biology, University of Antwerp, Antwerp, Belgium

*Correspondence to*: Yongshuo H. Fu (yfu@bnu.edu.cn), Yuyu Zhou (yuyuzhou@hku.hk)

**Abstract.** Global change has substantially shifted vegetation phenology, with important implications in the carbon and water cycles of terrestrial ecosystems. Various vegetation phenology datasets have been developed using remote sensing data; however, the significant uncertainties in these datasets limit our understanding of ecosystem dynamics in terms of phenology. It is therefore crucial to generate a reliable large-scale vegetation phenology dataset, by fusing various existing vegetation phenology datasets, to provide a comprehensive and accurate estimation of vegetation phenology with fine spatiotemporal resolution. In this study, we merged four widely used vegetation phenology datasets to generate a new dataset using the Reliability Ensemble Averaging fusion method. The new dataset has a spatial resolution of 0.05° and covers the period from 1982 to 2020, with geographic coverage extending above 30 degrees North in the Northern Hemisphere. The evaluation using ground-based phenocam data from 280 sites indicated that the accuracy of the newly merged dataset was substantially improved compared to the four original datasets. The start and end of the growing season in the newly merged dataset showed the highest correlation with ground-based phenocam observations, compared to the original datasets (0.84 and 0.71, respectively) and accuracy in terms of the root mean square error between phenocam data and merged datasets (12 and 17 d, respectively). Using the new dataset, we found that the start of the growing season is occurring approximately 0.19 days earlier per year (p < 0.01), while the end of the growing season is occurring 0.18 days later per year (p < 0.01) over the period 1982 – 2020 across regions north of 30° N. This dataset offers a unique and novel source of vegetation phenology data for global ecology studies.

## 1 Introduction

Global change has notably altered the timing of vegetation phenology (Ettinger et al., 2020; Zhang et al., 2022), leading to important impacts on the carbon and water cycles of terrestrial ecosystems (Peñuelas et al., 2009; Piao et al., 2019; Richardson et al., 2012; Zhou, 2022). Various vegetation phenology datasets using remote sensing data have been produced, but inconsistencies and uncertainties arise when comparing these datasets with ground-based phenological observations, and there is large variation in spatiotemporal resolution (Peng et al., 2017). Therefore, there is an urgent need to develop a highly reliable vegetation phenology product to improve our understanding of vegetation phenology dynamics, and to facilitate subsequent research on terrestrial ecosystem responses to climate change.

Ground-based phenological records are commonly used in vegetation phenology studies (Fu et al., 2014; Geng et al., 2020; Sparks and Carey, 1995; Zhou et al., 2020). For example, phenocams, a near-surface remote sensing tool, have been operational for more than 20 years (Richardson et al., 2018a). Although ground-based observations provide high accuracy in terms of phenology dynamics, they are limited to certain locations, resulting in sparse spatial coverage. In contrast, phenology datasets based on remote sensing data can cover large areas, providing comprehensive and continuous monitoring of vegetation phenology across landscapes, regions, or even continents. Additionally, remote sensing datasets and phenocam data are processed using standardized methods that ensure consistency and comparability across different locations and periods. However, phenology datasets based on remote sensing data do have certain limitations. Owing to differences in revisit cycles among satellites, together with sensor characteristics, sun-sensor geometry, and atmospheric conditions during imaging, substantial bias exists among the derived phenology datasets. For example, differences of >50 d in the start of the growing season (SOS) have been reported among different phenology datasets based on remote sensing data (Peng et al., 2017; Zhou et al., 2020). Additionally, substantial variations in the trends of vegetation phenology exist. For example, a recent study reported that the SOS was delayed by 0.17 d $yr^{-1}$ when based on the Global Inventory Modeling and Mapping Studies-3rd Generation (GIMMS 3g) dataset, whereas the SOS was advanced by 0.58 d $yr^{-1}$ when based on the Moderate Resolution Imaging Spectroradiometer (MODIS) dataset in the Northern Hemisphere during 2000－2015 (Zhang et al., 2020). Previous studies found that different vegetation phenology datasets have advantages and disadvantages in different regions and over different periods (Fensholt and Proud, 2012; Zhang et al., 2020). The MODIS phenology product for the United States shows a stronger correlation with ground observations compared to the AVHRR (Advanced Very High-Resolution Radiometer) phenology product (Peng et al., 2017), while the VIPPHEN (Making Earth System Data Records for Use in Research Environments Vegetation Index and Phenology) data has fewer missing values than the MODIS phenology product. For estimates obtained using different extraction methods, such as varying algorithms or approaches for extracting SOS and EOS (the end of the growing season) from the same satellite data, the discrepancies can exceed one month (White et al., 2009). Additionally, the NDVI (Normalized Difference Vegetation Index) is a commonly used remote sensing indicator for assessing vegetation cover and health. It is derived from reflectance data in the red and near-infrared bands and is closely related to

aboveground net primary productivity (Myneni et al., 1995; Pettorelli et al., 2005). The NDVI threshold required for phenology extraction varies across different biomes due to differences in vegetation types, growth patterns, and environmental conditions, which affect how NDVI values correspond to phenological events such as the start and end of the growing season (Reed et al., 1994). Because it is difficult to determine the optimal dataset from the various phenology datasets, producing a merged dataset using a method that selects the most suitable dataset for different times and locations from all input datasets is essential for providing a comprehensive and accurate estimation of vegetation phenology with high spatiotemporal resolution.

Remote sensing vegetation phenology typically reflects key transition dates in the vegetation growth cycle, such as the start and end of the growing season, using various vegetation indices such as NDVI (Normalized Difference Vegetation Index) and EVI (Enhanced Vegetation Index) to assess vegetation conditions (Cong et al., 2012; Piao et al., 2019). Ideally, phenological dates extracted from different methods (thresholds, derivatives, smoothing functions, and fitted models) should accurately capture changes in actual physiological conditions (De Beurs and Henebry, 2010). However, in existing phenology datasets, there is often no "best" definition of these transition dates (White et al., 2009). The effectiveness of different extraction methods can vary across regions and periods, and may not always perfectly reflect true vegetation conditions (Cong et al., 2013; De Beurs and Henebry, 2010). For instance, in high-latitude areas, meaningful observations are relatively sparse. If the smoothing method removes too much information, it may reduce the ability to extract phenological signals that accurately reflect surface dynamics (Wang et al., 2015). Therefore, integrating multiple datasets based on their reliability, rather than relying on a single dataset or using a simple averaging method, is a more robust approach. Our study stresses the value of integrating different datasets, and phenology results can be improved by applying the REA method, which assigns weights based on dataset reliability. This method can reduce uncertainties and provide a more accurate representation of phenological dynamics across different spatial and temporal scales.

Data fusion methods generally include unmixing-based, weight-function-based, and Bayesian-based approaches (Gevaert and García-Haro, 2015; Piao et al., 2019). In vegetation phenology studies, fusion methods based on raw remote sensing data, such as the Spatial and Temporal Adaptive Reflectance Fusion Model (Gao et al., 2006) and the Enhanced Spatial and Temporal Adaptive Reflectance Fusion Model (Zhu et al., 2010), are often influenced by vegetation types, growth conditions, and methodological assumptions (Sisheber et al., 2022). These methods are typically applied to specific regions, and their performance can be affected by nonlinear spectral mixing, where the reflectance of vegetation endmembers (the pure spectral signatures of distinct land cover types) changes nonlinearly, and the spectral response of a single pixel is no longer a simple linear combination of the endmember spectra (Ma et al., 2015). The nonlinear combination of the ground feature can degrade the accuracy of vegetation phenology extraction. Unlike these approaches, the Reliability Ensemble Averaging (REA) method is not based on the assumption of linear reflectance changes. Instead, it directly merges annual phenology products based on their reliability. Compared to traditional data fusion methods, the REA method shows clear advantages in simplicity and computational efficiency (Lu et al., 2021) while explicitly accounting for dataset reliability, in contrast to simple averaging methods that assume equal reliability across datasets. The simple averaging method treats all datasets equally by calculating

the mean value across different vegetation phenology datasets, despite their uncertainties varying across time and space (Lu et al., 2021; Wang et al., 2019), which leads to potential inaccuracies in the result. The REA method considers the temporal consistency of vegetation phenology data and uses the voting principle, whereby the final REA result is generated by assigning different weights to the input data sources based on their agreement (Giorgi and Mearns, 2002), which provides convergence while preserving spatial differences, making it suitable for multi-source data fusion.

In this study, we merged four widely used vegetation phenology datasets to generate a new dataset using the REA fusion method. The spatial resolution of the new dataset is 0.05°, with a temporal scale spanning 1982‑2020, and it covers regions north of 30° N latitude. The new dataset was evaluated using data from the ground-based phenocam dataset from 280 sites over the period 2000–2018, which provided 1410 site-year combinations. We further explored the phenological trends in spring and autumn vegetation phenology using the merged dataset. The new vegetation phenology dataset could be used in further studies on the impact of energy and carbon-water cycles within terrestrial ecosystems, together with analysis of their responses and feedbacks to global climate change (Piao et al., 2009, 2019; Tang et al., 2016).

## 2 Data and Method

### 2.1 Phenology dataset

Four satellite-based vegetation phenology products were used to create a merged dataset, and the ground-based phenocam dataset was used for validation. The four satellite-based vegetation phenology products include (1) the MCD12Q2 (Moderate Resolution Imaging Spectroradiometer (MODIS) Land Cover Dynamics data product) phenology dataset, which was extracted from the MODIS Land Cover Dynamics Version 6.1 derived by Friedl et al., 2022; (2) the VIP (Making Earth System Data Records for Use in Research Environments Vegetation Index and Phenology) dataset derived by Didan and Barreto, 2016, (3) the GIM_3g (GIMMS (Global Inventory Modeling and Mapping Studies) Normalized Difference Vegetation Index 3rd Generation) based phenology dataset derived by Wang et al. (2019), and (4) the GIM_4g (GIMMS(Global Inventory Modeling and Mapping Studies) Normalized Difference Vegetation Index 4th Generation) based phenology dataset derived by Chen and Fu, 2024. These phenological data products were obtained from open sources and used to merge a new set of phenological products. The time span and the spatial resolution of each vegetation phenology dataset are listed in Table 1. The merged data used in this paper were clipped into regions above 30 degrees in the Northern Hemisphere to ensure that the region was covered by four datasets.

**Table 1 List of data sources**

| Name | Abbreviations | Sensor | Spatial Resolution | Time Span | Reference |
|---|---|---|---|---|---|
| MODIS MCD12Q2 | MCD12Q2 | MODIS | 500m | 2001–2020 | (Friedl et al., 2022) |

| MEaSUREs VIPPHEN | VIP | AVHRR& MODIS | 0.05° | 1982–2015 | (Didan and Barreto, 2016) |
|---|---|---|---|---|---|
| GIMMS NDVI3g | GIM_3g | AVHRR | 1/12° | 1999–2014 | (Wang et al., 2019) |
| GIMMS NDVI4g | GIM_4g | AVHRR | 1/12° | 1982–2020 | (Chen and Fu, 2024) |

**2.1.1 MCD12Q2 phenology dataset**

The MCD12Q2 product was derived using data from the MODIS sensor onboard the Terra and Aqua satellites. The MCD12Q2 land cover dynamic product v6.1 provides a global surface phenology dataset with a 500-m spatial resolution for the period 2001–2020. The vegetation phenology data were extracted from the Nadir Bidirectional Adjusted Reflectance 2-band Enhanced Vegetation Index (EVI2) using the threshold method (Gray et al., 2019). The threshold method defines the growing state of the vegetation as the time when the vegetation index reaches a certain percentage of the annual amplitude and reflects a specific vegetation physiological growth stage (the schematic diagram is shown in Figure S1). The amplitude is calculated as the difference between the maximum and minimum values of the EVI2 time series during the entire growing season. The MCD12Q2 phenology dataset includes greenup and dormancy (equivalent to SOS and EOS in this study, respectively). Greenup (dormancy) is defined as the date when the EVI2 time series first (last) crosses 15% of the segment EVI2 amplitude (Gray et al., 2019). The time series data were fitted by a penalized cubic smoothing spline. This dataset can be found at https://lpdaac.usgs.gov/products/mcd12q2v061/ (Friedl et al., 2022).

**2.1.2 VIP phenology dataset**

The VIP phenology dataset was generated using data from the NASA Making Earth System Data Records for Use in Research Environments (MEaSUREs) and the Advanced Very High-Resolution Radiometer (AVHRR) over the period 1981–1999, together with MODIS/Terra MOD09 surface reflectance data over the period 2000‒2014 (Didan et al., 2018). The VIP dataset includes the SOS and EOS, which were also extracted using the threshold method. The filtering method, which uses confidence intervals and an operational continuity algorithm, was applied to reconstruct the time series curves. The start (end) of the season is defined using the modified Half-Max method (White et al., 2009) as the date when the NDVI time series first (last) crosses 35% of the growing season NDVI amplitude. This dataset is organized in a geographic gridded format with a spatial resolution of 0.05°. This dataset can be found at https://lpdaac.usgs.gov/products/vipphen_ndviv004/ (Didan and Barreto, 2016).

**2.1.3 GIM_3g phenology dataset**

The GIMMS NDVI 3g-based phenology dataset (GIM_3g) has a spatial resolution of 1/12° and covers the period 1999–2014 (Wang et al., 2019). A double logistic function was applied to fit the NDVI curve and the threshold method was used to extract phenological dates, including the SOS and EOS. This product provides phenology data for the Northern Hemisphere, and it uses the date when the NDVI first (last) crosses 20% of the segment NDVI amplitude as the SOS (EOS). This dataset

can be accessed at http://data.globalecology.unh.edu/data/GIMMS_NDVI3g_Phenology/ (Wang et al., 2019).

### 2.1.4 GIM_4g phenology dataset

The GIM_4g dataset, based on the GIMMS NDVI 4g dataset acquired by the AVHRR sensors, has a spatial resolution of 1/12° and a temporal scale spanning 1982–2020. Two steps were adopted in the process to extract phenological dates. First, the NDVI time series data were fitted and smoothed using five fitting methods: the HANTS-Maximum, Spline-Midpoint, Gaussian-Midpoint, Timesat-SG, and Polyfit-Maximum methods. Second, the threshold method was used to extract phenological dates, using the date when the NDVI first (last) crosses 20% (50%) of the segment NDVI amplitude as the SOS
(EOS) (Chen et al., 2024; Chen and Fu, 2024). The average spring (SOS) and autumn (EOS) phenological dates were produced by averaging the results obtained from the five fitting methods. The GIM_4g phenology dataset is available at https://doi.org/10.5281/zenodo.11136967 (Chen and Fu, 2024).

### 2.1.5 Camera-based phenology dataset

A ground-based phenocam dataset, with phenological dates extracted from camera-derived images with high spatial
resolution and reliable accuracy, was used to validate the merged dataset. The phenocam dataset comprises three datasets. The first dataset, the PhenoCam Dataset v2.0 (Richardson et al., 2018b; Seyednasrollah et al., 2019a, b), includes data derived from conventional visible-wavelength automated digital camera imagery through the PhenoCam Network (Richardson et al., 2018a) over the period 2000–2018 and across 393 sites in various ecosystems. For detailed information, please refer to https://daac.ornl.gov/VEGETATION/guides/PhenoCam_V2.html and https://phenocam.nau.edu/webcam/. It comprises all
typical vegetation types including deciduous broadleaf, deciduous needleleaf, evergreen broadleaf, evergreen needleleaf, grassland, mixed vegetation, shrubland, tundra, and wetland ecosystems, mainly in regions of Europe and North America (Moon et al., 2021; Ruan et al., 2023). A spline interpolation method was applied to the PhenoCam data to extract transition dates for each region of interest (ROI) using the green chromatic coordinate (GCC), a measure of greenness intensity derived from digital imagery (Sonnentag et al., 2012) in the PhenoCam Dataset v2.0. We used the date when the GCC first (last) crosses
25% of the GCC amplitude as the SOS and EOS. The second phenocam dataset is from the Japan Internet Nature Information System digital camera data (http://www.sizenken.biodic.go.jp/) acquired over the period 2002–2009. Ide and Oguma(Ide and Oguma, 2010) provided greenup dates for two phenocam sites with the ROI defined at the species level scale. The vegetation types included in their data comprised wetland and mixed deciduous forest. The date of green-up each year was estimated as the day of year (DOY) corresponding to the maximum rate of increase in the 2G-RBi index. This is calculated as the maximum
of the second derivative of GCC time series (the maximum of the second derivative of GCC) The third dataset consists of phenology data for deciduous broadleaf forests in Japan (Inoue et al., 2014), supported by the Phenological Eyes Network (http://www.pheno-eye.org/), which is a network of ground-based observatories for long-term automatic observation of vegetation dynamics established in 2003 (Nasahara and Nagai, 2015), the start and end of season is defined as the first day when 20% of leaves had flushed and the first day when 80% of leaves had fallen in the given ROI, respectively. For this study,

we excluded 26 sites that only provided one type of transition date (either SOS or EOS) and removed 90 sites where none of the four remote sensing datasets provided valid phenology estimates. These excluded sites were primarily located in cropland-dominated areas or regions with sparse vegetation, where the low spatial resolution of remote sensing data limits the reliable detection of phenological transitions. We then selected phenocam data from 280 sites over the period 2000 – 2018, resulting in 1410 site-year combinations.

### 2.1.6 Land cover dataset

To avoid the impact of human activities and non-vegetated areas on data quality, areas of cropland, cropland/natural vegetation mosaics, permanent snow and ice, barren land and water bodies were removed based on a land cover dataset obtained by supervised classification of MODIS reflectance data (Sulla-Menashe and Friedl, 2018). The land cover data generated based on the Annual International Geosphere–Biosphere Programme classification schemes, are available from
https://lpdaac.usgs.gov/products/mcd12q1v061/ (Friedl and Sulla-Menashe, 2022).

### 2.2 Ensemble method for estimating phenological dates

### 2.2.1 Reliability ensemble averaging method

The weighting method was applied to obtain more accurate SOS and EOS dates from the four vegetation phenology datasets. The weight assigned to each product was based on the interannual variability of each phenology dataset, as well as
the degree of consistency and offset among the four datasets. Importantly, these weights can vary over time to reflect changes in dataset reliability and performance (Giorgi and Mearns, 2002). The consistency is measured as the difference between each input dataset and the mean value of the four datasets, and the offset is measured as the difference between the REA result and each input dataset. These values are iteratively calculated during the process of determining the final weight coefficients. There are discrepancies in the spatial coverage among the four phenology datasets, and missing data occurs in specific regions for
some of the datasets. The ensemble method can fill in missing data accurately, thereby producing a phenology dataset with high accuracy and spatially continuous coverage. Furthermore, the process of merging the phenology datasets does not depend on simple averaging; instead, it is based on the uncertainty (calculated using merged result and the differences between the REA result and the remote sensing phenology datasets) among the products, which produces data that is more reliable than those obtained using the simple averaging method, and can circumvent the effects of outliers (Giorgi and Mearns, 2002).

The REA method, based on the 'voting principle' (where the REA result is generated by assigning different weights to the input data sources), produces data that aligns with the majority of input phenology products at the pixel level. This approach assumes that most data values at each pixel are accurate, while outliers are down-weighted or excluded. It provides a dataset with high reliability by relying on the temporal consistency of each pixel among the input products, and by minimizing the influence of outliers during the merging process (Giorgi and Mearns, 2002). The REA method has been applied to generate
datasets for multiple elemental fields, e.g., temperature, evapotranspiration, and precipitation (Giorgi and Mearns, 2002; Lu et

al., 2021; Xu et al., 2010). In this study, the REA method was used to integrate both the SOS and the EOS from the four phenology datasets.

The REA method gives different weights to the various datasets involved in the process of data merging, and then obtains the desired result using the following function:

$$\widetilde{\Delta Phe} = \tilde{A}(\Delta Phe) = \frac{\sum_i R_i \Delta Phe_i}{\sum_i R_i} \tag{1}$$

where $\widetilde{\Delta Phe}$ represents the phenology result, $\Delta Phe_i$ represents the different datasets involved in the process, $\tilde{A}$ denotes the REA process, and $R_i$ represents the model reliability factor, which is defined as follows:

$$R_i = \left[ \left(R_{B,i}\right)^m \times \left(R_{D,i}\right)^n \right]^{\left[\frac{1}{m \times n}\right]}$$
$$= \left\{ \left[ \frac{\epsilon_{Phe}}{abs(B_{Phe,i})} \right]^m \left[ \frac{\epsilon_{Phe}}{abs(D_{Phe,i})} \right]^n \right\}^{\left[\frac{1}{m \times n}\right]} \tag{2}$$

where $R_{B,i}$ measures the bias of the data compared with that of the average data (the higher the bias, the lower the reliability of the dataset), and $R_{D,i}$ represents the convergence criterion of the data (the larger the distance between the dataset and the newly generated REA data, the poorer the convergence; several iterations are required to reach convergence). The values of $R_{B,i}$ and $R_{D,i}$ will be set to 1 when $B_{Phe,i}$ and $D_{Phe,i}$ are less than $\epsilon_{Phe}$, which means the deviation of the dataset is within the limit of natural variation.

$$B_{Phe,i} = \Delta Phe_i - \overline{Phe}, \tag{3}$$

$$D_{Phe,i} = \Delta Phe_i - \Delta Phe, \tag{4}$$

$$\varepsilon_{Phe} = \max\big(MA(D_{Phe})\big) - \min\big(MA(D_{Phe})\big). \tag{5}$$

Equation (3) explains the derivation of $B_{Phe,i}$, it is defined by the difference between the input dataset and the mean value of the four datasets. Equation (4) explains the arithmetic process of $D_{Phe,i}$, which is measured by the difference between the REA result and each input dataset. In Eq. (5), $\varepsilon_{Phe}$ is measured by the natural variability in phenology, which is calculated by estimating the difference between the maximum and minimum values of the multiyear moving averages following linear detrending of the observed long-term series data, and works with $B_{Phe,i}$ and $D_{Phe,i}$ jointly to assign weights to each dataset. Natural variability changes from region to region, in Equation (1) and (6), $\varepsilon_{Phe}$ cancels out under the condition of $B_{Phe,i}$ and $D_{Phe,i}$ greater than $\varepsilon_{Phe}$, which is based on the assumption that more stringent are required to increase the reliability over regions characterized by lower natural variability.

$$\delta_{Phe} = \left[ \tilde{A}\big(\Delta Phe_i - \widetilde{\Delta Phe}\big)^2 \right]^{\frac{1}{2}} = \left[ \frac{\sum_i R_i \big(\Delta Phe_i - \widetilde{\Delta Phe}\big)^2}{\sum_i R_i} \right]^{\frac{1}{2}} \tag{6}$$

$$\Delta Phe_+ = \widetilde{\Delta Phe} + \tilde{\delta}_{\Delta Phe}, \tag{7a}$$

$$\Delta Phe_- = \widetilde{\Delta Phe} - \tilde{\delta}_{\Delta Phe}. \tag{7b}$$

In Eq. (6), $\delta_{Phe}$ is the uncertainty range calculated using $R_i$ and the difference between the REA result and the datasets (a higher value of $\delta_{Phe}$ means larger differences between the REA result and the original phenology datasets). The upper and lower uncertainty limits are measured by $\widetilde{\Delta Phe}$ and $\tilde{\delta}_{\Delta Phe}$, respectively, in Eqs. (7a) and (7b).

If an individual data point shows significant discrepancies compared to others, potentially caused by improper extraction methods in that region, the $B_{Phe,i}$ and $D_{Phe,i}$ will extract this variance and incorporate it, along with the natural variability $\varepsilon_{Phe}$ of the region into the weight distribution process. If the natural variability of that region is low, then the weight is assigned to a smaller value, and if the natural variability of the region is large, the weight is assigned by both the natural variability and the deviations.

To evaluate the robustness of the REA method, we used sensitivity analysis to confirm its reliability, considering the influence of different time spans and dataset combination choices on the fusion results respectively. In the sensitivity analysis of the time-span, we performed fusion experiments using two subsets (2001–2005 and 2006–2010) and compared them to the full 2001–2010 period, and the differences in the fusion results were analyzed. In the sensitivity analysis of the selection of different dataset combinations on the fusion results, we selected three datasets combinations for the fusion experiment and analyzed the influence of data selection on the fusion results. Each group included two datasets selected from the four used in the full REA fusion. These combinations were chosen to assess how different data sources influence the final REA results.

### 2.2.2 Evaluation criteria

In this study, the metrics of the root mean square error (RMSE), BIAS, correlation coefficient (r), unbiased RMSE (UbRMSE) and coefficient of variation (CV) were used for data evaluation:

$$RMSE = \sqrt{\frac{\sum_{i=1}^{n}(Phe_i - Ref_i)^2}{n}}, \tag{8}$$

$$BIAS = \frac{\sum_{i=1}^{n}(Phe_i - Ref_i)}{n}, \tag{9}$$

$$r = \frac{\sum_{i=1}^{n}\left(Phe_i - \overline{Phe}\right)\left(Ref_i - \overline{Ref}\right)}{\sqrt{\sum_{i=1}^{n}\left(Phe_i - \overline{Phe}\right)^2}\sqrt{\sum_{i=1}^{n}\left(Ref_i - \overline{Ref}\right)^2}}, \tag{10}$$

$$ubRMSE = \sqrt{RMSE^2 - BIAS^2}, \tag{11}$$

$$STD = \sqrt{\frac{1}{N}\sum_{i=1}^{N}\left(Phe_i - \overline{Phe}\right)^2}, \tag{12}$$

$$CV = \frac{\sigma_{Phe}}{\overline{Phe}} \tag{13}$$

where $n$ represents the number of site years, $Phe_i$ represents the corresponding vegetation phenological indicator (SOS and EOS) at a given point, $Ref_i$ represents data from a phenology camera, $\sigma_{Phe}$ represents the standard deviation of $Phe_i$, and $\overline{Phe}$ and $\overline{Ref}$ represent the average of $Phe_i$ and $Ref_i$, respectively.

RMSE is calculated as the square root of the average of the squares of the residuals, which penalizes larger errors than smaller ones and provides an estimate of the magnitude of errors between remote sensing estimated value and phenocam datasets. BIAS is the average difference between the remote sensing estimated value and the phenocam value, which helps in understanding whether the estimated value is higher or lower than the phenocam value. The correlation coefficient measures the linear relationship between two variables. The ubRMSE measures the deviation between two variables without systematic errors. Standard deviation quantifies the variation of the dataset, which measures the deviation between data and the mean value.

### 2.2.3 Mann–Kendall trend test

The Mann–Kendall trend test is a nonparametric trend test method, which has the characteristics of not being limited by a specific distribution and a small number of outliers, and can be used to detect the trend of time series data (Kendall, 1975). The Mann-Kendall test was applied to detect trends in SOS and EOS dates during 1982－2020 in the merged dataset, as well as trends in growing season greenness across different phenology datasets. The SOS/EOS trend refers to the temporal change in the timing of phenological events and the greenness trends refer to interannual variations in vegetation greenness during the growing seasons. The basic Mann–Kendall test formulas are as follows:

$$S = \sum_{i=1}^{n-1} \sum_{j=j+1}^{n} \text{sgn}(X_j - X_i), \tag{14}$$

$$Z_c = \begin{cases} \dfrac{S-1}{\sqrt{Var(S)}} & S > 0 \\ 0 & S = 0 \\ \dfrac{S+1}{\sqrt{Var(S)}} & S < 0 \end{cases} \tag{15}$$

where $X_i$ and $X_j$ are the phenological parameter values of the $i$-th year and the $j$-th year of the pixel, respectively, $n$ is the length of the time series, $sgn$ is the sign function, and $S$ is the test statistic. The null hypothesis $H_0$: the time series data is $n$ independent samples with identically distributed random variables, $H_1$: for any $i, j \leq n$, and $i \neq j$, the distribution of $X_i$, $X_k$ is different. If $|Z| \geq Z_{1-\frac{\alpha}{2}}$, the time series is considered to have a statistically significant change; otherwise, any change is considered not statistically significant. When $Z > 0$, the time series has an upward trend; when $Z < 0$, it has a downward trend (Zhou and Liu, 2018).

### 2.2.4 Growing season greenness

Greenness is a widely used indicator of vegetation growth, typically represented by NDVI (Myneni, 1997). The Growing

Season Greenness (GSG) is calculated as the mean NDVI value within the growing season in this study, defined by the period between SOS and EOS:

$$GSG = Mean(NDVI[Date_{SOS}, Date_{EOS}]) \tag{16}$$

$Date_{SOS}$ is the day of year value of SOS data, and $Date_{EOS}$ is the day of year value of EOS value, $Mean$ denotes calculating the mean NDVI value in the corresponding date range.

## 3 Results

### 3.1 Difference in vegetation phenological dates among the four datasets

Figure 1 illustrates the spatial distribution of the multiyear mean dates for both the SOS and the EOS above 30°N for each of the four datasets. The mean SOS values for the MCD12Q2, VIP, GIM_3g, and GIM_4g datasets are day of the year (DOY) 120 (std = 32 d), 125 (std = 43 d), 132 (std = 17 d), and 139 (std = 32 d), respectively. Discrepancies among the datasets are particularly notable in southwestern North America, North Africa, the Qinghai–Tibet Plateau, and Mongolia. Compared with the SOS, the EOS exhibits greater variability, and the mean EOS values for the MCD12Q2, VIP, GIM_3g, and GIM_4g datasets are DOY 281 (std = 37 d), 290 (std = 44 d), 315 (std = 19 d), and 287 (std = 53 d), respectively. Among the four datasets, the spatial distributions of the GIM_4g and VIP datasets are the most similar. In comparison with these two datasets, the MCD12Q2 dataset displays earlier EOS values in Northern Europe, Central Asia, North America, and the 45°–60°N latitudinal belt over Central Asia. Given the substantial differences among these datasets, it is imperative to integrate these datasets into a merged dataset with higher accuracy.

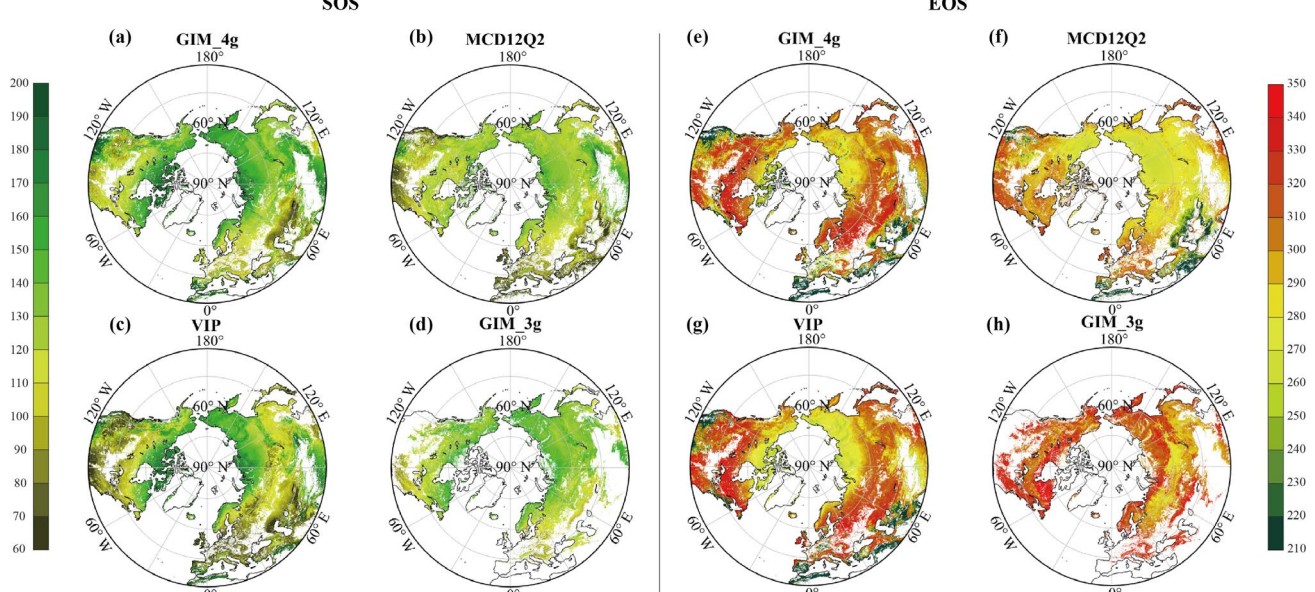

**Figure 1: Spatial distribution of multiyear mean SOS and EOS dates from each phenology dataset:** (a–d) multiyear mean SOS dates and (e–h) multiyear mean EOS dates derived from the GIM_4g, MCD12Q2, VIP, and GIM_3g datasets, respectively.

**3.2 Variation of weights and contributions of the four datasets to the merged phenology dataset**

The weight of each dataset, as determined by the REA method, varies largely among years and specific locations. The left panels of Fig. 2 illustrate the mean weight for each dataset in each year over the period 1982–2020, with the upper and lower sections representing the SOS and the EOS, respectively. For the SOS, the overall weight of the VIP dataset during 1982–1998 surpasses that of the GIM_4g dataset. The GIM_3g dataset is dominant during 1999–2014, with weights exceeding 65%. In 2015, the weighting of the MCD12Q2 dataset was highest at approximately 45%, with the weights of the 310    other two datasets broadly similar. During 2016–2020, the weights of the MCD12Q2 and GIM_4g datasets were 61% and 39%, respectively. The combinations of data sources for the EOS data are similar to those for the SOS data. Specifically, during 1982–1998, the weight of the VIP dataset was approximately 65%, with the GIM_4g dataset accounting for the remaining 35%. For 1999–2000, the weighting of the GIM_3g dataset is approximately only 10%, whereas that of the VIP dataset is the highest (approximately 55%). Throughout the period 2001-2014, the weighting of the VIP dataset is greatest (>45%), whereas 315    that of the GIM_3g dataset is low (<10%); the weighting of the GIM_4g and MCD12Q2 datasets each account for over 20%. During 2016–2020, the weights of the GIM_4g and MCD12Q2 datasets are broadly equal, albeit with the weighting of the GIM_4g dataset slightly exceeding that of the MCD12Q2 dataset.

    The latitudinal distribution of the mean weighting of the datasets for the SOS and the EOS is shown in Fig. 2(b) and 2(d), respectively. For the SOS data, the zonal distribution of the GIM_4g, VIP, and MCD12Q2 datasets is reasonably stable within 320    30°–75°N. The weight of the GIM_3g dataset is notably higher between 50°N and 70°N, primarily because of its spatial distribution, and it shows notable fluctuations in high-latitude areas. In contrast, the weighting of the EOS datasets exhibits relatively smooth changes within 30°–75°N. There are marked fluctuations in the weighting of the GIM_4g and VIP datasets in high-latitude areas above 75°N. The weight of the GIM_4g dataset between 30°N and 75°N fluctuates before stabilizing smoothly. Conversely, the weight of the VIP dataset increases with latitude, displaying a trend opposite to that of the GIM_4g 325    dataset. Additionally, the weighting of both the MCD12Q2 and the GIM_3g datasets initially increases and then decreases with increasing latitude.

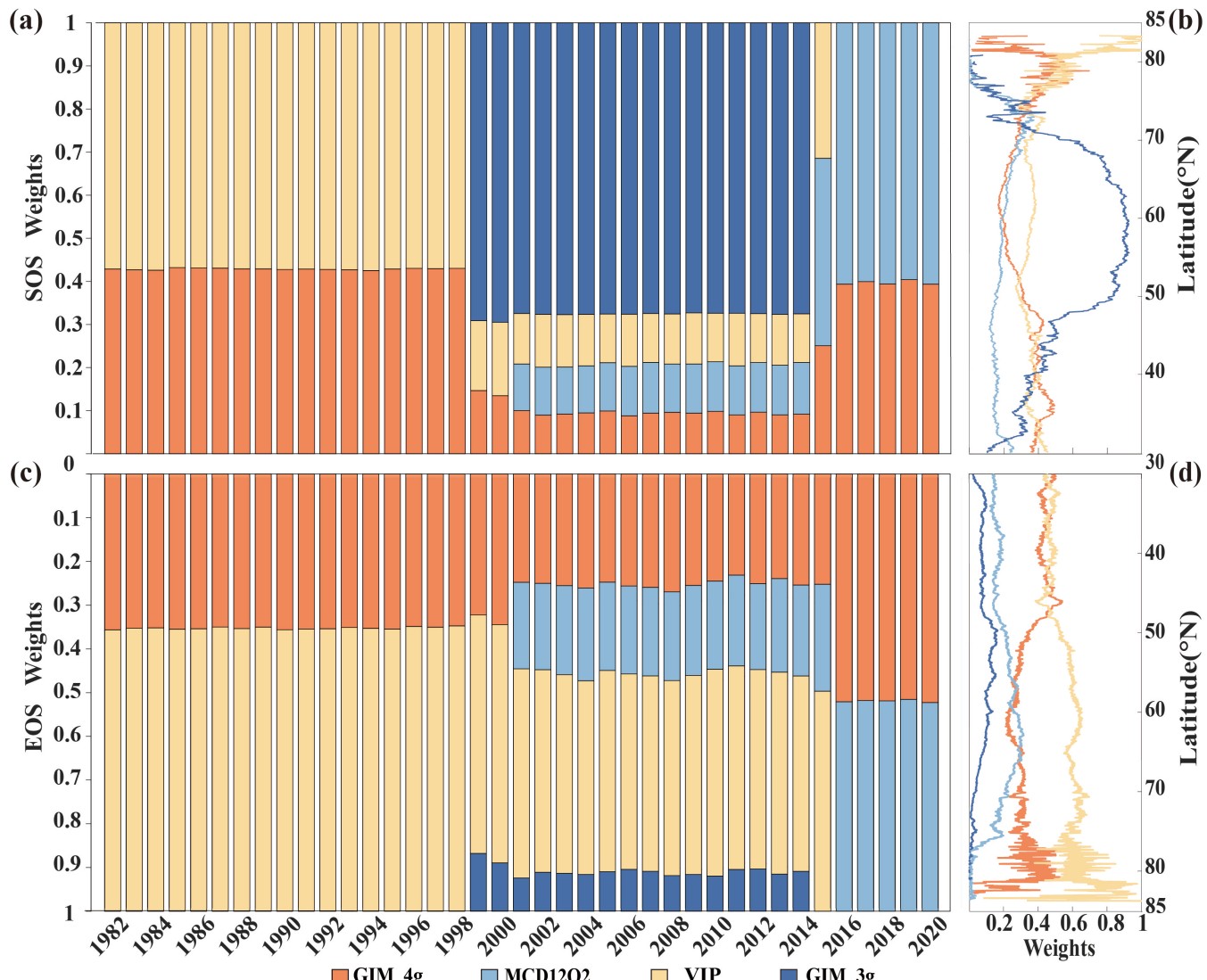

**Figure 2: (a and c) Weights of the four phenology datasets during 1982－2020 and (b and d) latitudinal differences for (upper) the SOS and (lower) the EOS.** For panels b and d, the latitudinal weights are the mean values of each dataset over their respective timespans. The four datasets comprise the GIM_4g, MCD12Q2, VIP, and GIM_3g datasets (for the full names, see Table 1).

Figure 3 shows the spatial distribution of the mean contribution of the four datasets to the merged SOS and EOS results, calculated as the average weight for each pixel over the timespan for the corresponding dataset. For the SOS data, the GIM_3g dataset exhibits the greatest contribution, followed by similar contributions from the GIM_4g and VIP datasets; the MCD12Q2 dataset has the smallest contribution. The MCD12Q2 dataset has a greater contribution in high-latitude areas near the Arctic Circle, but makes a smaller contribution in most other regions. The VIP dataset generally has a greater contribution than that of the MCD12Q2 dataset, with values ranging between 0 and 0.5 in 90% of areas. The overall contribution of the GIM_3g dataset is reasonably uniform, averaging at approximately 0.37. For the EOS data, the VIP dataset has the greatest contribution,

followed by the GIM_4g dataset; the GIM_3g dataset has the smallest contribution. The contribution of the MCD12Q2 dataset remains relatively small, primarily distributed between 0 and 0.5. The VIP dataset has a positive correlation with latitude, with approximately 4.7% of areas of weights exceeding 0.8 in central Asia and parts of East Asia, whereas the contribution of the GIM_3g dataset remains lower across the entire region.

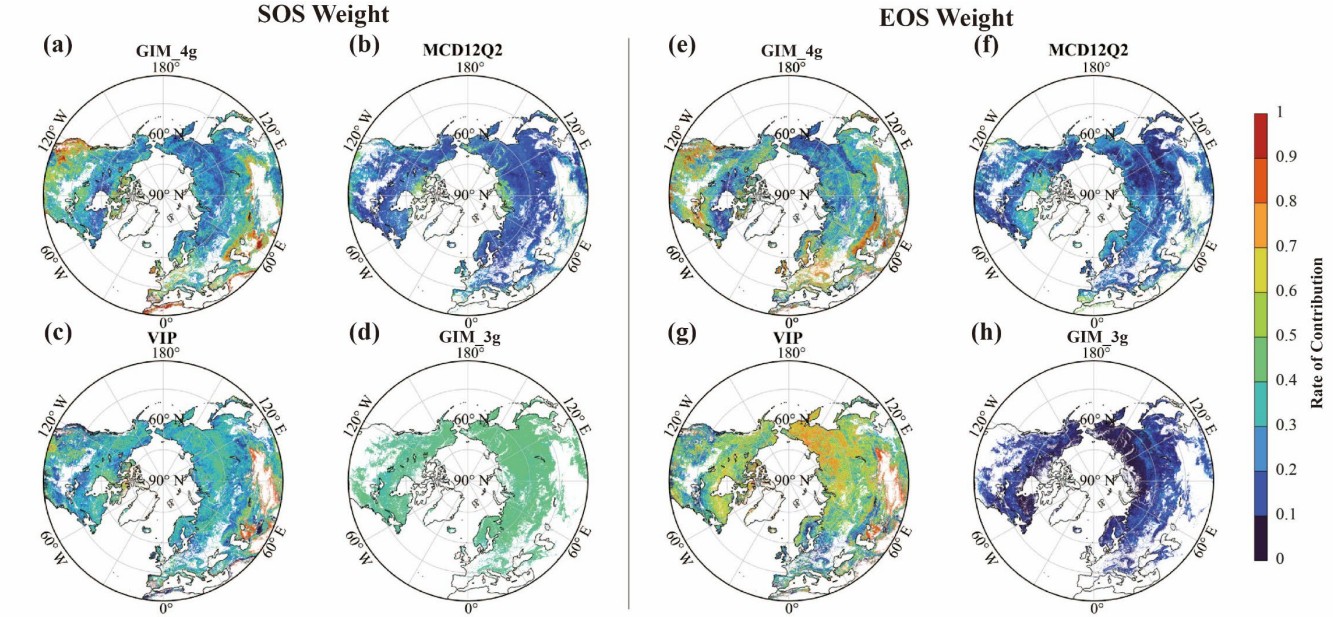

**Figure 3: Spatial distribution of the mean contribution of the four datasets to the merged SOS and EOS results.** The mean SOS (a-d) and EOS (e-h)weight derived from the GIM_4g, MCD12Q2, VIP, and GIM_3g datasets.

## 3.3 Merged phenology dataset using the REA method

Figure 4 displays the merged mean SOS and EOS dates for the period 1982–2020. For the SOS, a general pattern of increase with latitude is evident, albeit with a later occurrence of the SOS in southwestern North America and on the Qinghai–Tibet Plateau. The highest proportion of the SOS falls within DOY 120–150 (38.2%), followed by DOY 90–120 (23.1%), DOY 150–180 (21.23%), and DOY 60–90 (11.1%). Only a small proportion (4.6%) of areas are experiencing the SOS later than DOY 180. The mean SOS obtained using the REA method is DOY 129 (std = 28 d). It demonstrates an overall increase in the EOS with latitude, with fewer trends observed in high-latitude areas above 60°N and eastern parts of North America. The distribution of the EOS appears more uniform after merging. Unlike the SOS dates, which exhibit greater variability, the EOS dates are more consistent, predominantly occurring within day of year (DOY) 270–330 (80.0%). The mean EOS is DOY 284 (std = 20 d). Interannual variability in most regions for both the SOS and the EOS data is minimal; however, notable variations are observed in areas such as southwestern North America, Spain, Portugal, North Africa, West Asia, and Mongolia, consistent with the earlier analysis of data sources(Fu et al., 2014; Liu et al., 2016; Piao et al., 2006, 2015).

The mean uncertainty range of merged SOS and EOS dates, calculated using Equation (6), is presented in Figure 4. This range was determined using the REA method over the period from 1982 to 2020. The mean uncertainty range of SOS (EOS) dates is below 10d in more than 96% (94%) of regions, with less than 4% (5%) of regions exhibiting a mean uncertainty range exceeding 10d or 15d (Fig. 4b, e). The mean uncertainty range of SOS dates shows a negative correlation with latitude, whereas this trend is not evident in EOS dates. In Fig. 4(c, f), the coefficient of variation (CV) of the uncertainty in SOS and EOS dates from 1982 to 2020 were analyzed. More than 56% (73%) of regions exhibit a CV below 1, 31% (18%) of regions have a CV between 1 and 1.5, and only 13% (8%) of regions show a CV higher than 1.5. No evident correlation is observed between CV and latitude changes.

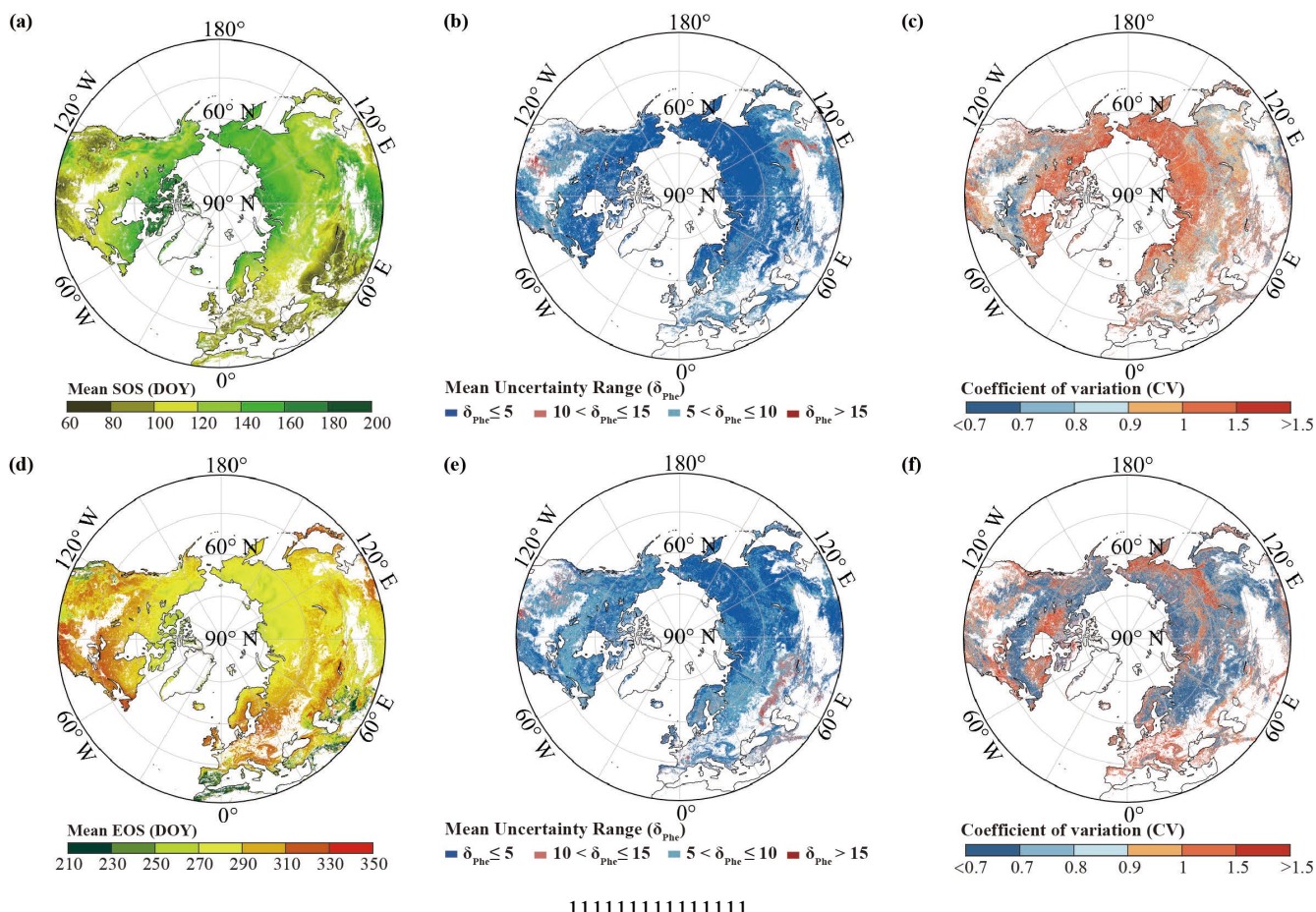

111111111111111

**Figure 4: Merged mean (a) SOS and (d) EOS dates (DOY) obtained using the REA method for the period 1982–2020 and the uncertainty in the REA merged data.** Mean uncertainty ($\delta_{Phe}$) of SOS dates (b) and EOS (e) obtained using the REA method for the period 1982–2020, and its coefficient of variation (CV) in merged SOS (c) and EOS dates (f).

The phenocam dataset was used to evaluate each of the four vegetation datasets and the merged dataset. Verification results of the SOS and EOS data indicate that the merged data produced using the REA method has the best performance (Fig. 5). Specifically, the RMSE for the SOS and the EOS is 12 and 17 d, respectively. The correlation between the SOS and

phenocam results is notably high at 0.84; for the EOS, it is 0.71. Evaluation of the four satellite-based SOS products shows that the GIM_3g dataset has the highest correlation coefficient and the lowest RMSE among the four datasets. However, it has

375 more missing values spatially and a shorter time span, leading to fewer points for verification. The MCD12Q2 dataset has a correlation coefficient of 0.65 and an RMSE of 20 d, but its wider spatial coverage provides more points for verification. The GIM_4g dataset has a lower correlation with the phenocam dataset owing to outliers, resulting in an RMSE of 29 d. Compared with the phenocam dataset, the VIP dataset tends to estimate earlier SOS dates than the other datasets, especially in areas where SOS occurs early in the season (DOY 100–140), leading to a larger RMSE. And comparing with simple average, the REA-

380 based SOS shows better performance in RMSE (REA and Average, 12d and 21d (in Figure S4), respectively), CORR (correlation coefficient) (REA and Average, 0.84 and 0.65, respectively), BIAS (REA and Average, -1.5d and -9.7d, respectively) and UbRMSE (REA and Average, 12d and 18d, respectively). The REA-based SOS dataset outperforms in terms of all indicators, with the lowest RMSE, UbRMSE, and standard deviation, together with the highest correlation and lowest absolute bias, thereby demonstrating high consistency with the phenocam dataset.

In the evaluation of the EOS, the MCD12Q2 dataset has the best results among the four datasets, and except for the REA result, it has the highest correlation coefficient and the lowest RMSE. The GIM_4g dataset shows good performance but tends to overestimate the EOS, with predicted dates occurring later than observed, resulting in an RMSE of 43 d. Both the VIP and the GIM_3g datasets also overestimate the EOS due to their spatial and temporal distributions, with RMSEs of 46 and 35 d, respectively. The REA-based EOS also shows better performance compared to the simple average of the original datasets in

RMSE (REA and Average, 17d and 32d, respectively), CORR (REA and Average, 0.71 and 0.45, respectively), BIAS (REA and Average, 1.0d and 8.0d, respectively) and UbRMSE (REA and Average, 17d and 31d, respectively). It is evident from Fig. 5 that the REA dataset demonstrates the highest accuracy and best consistency with the phenocam dataset, outperforming the four other datasets in terms of all indicators, with the lowest RMSE, UbRMSE, and standard deviation, together with the highest correlation and lowest absolute bias.

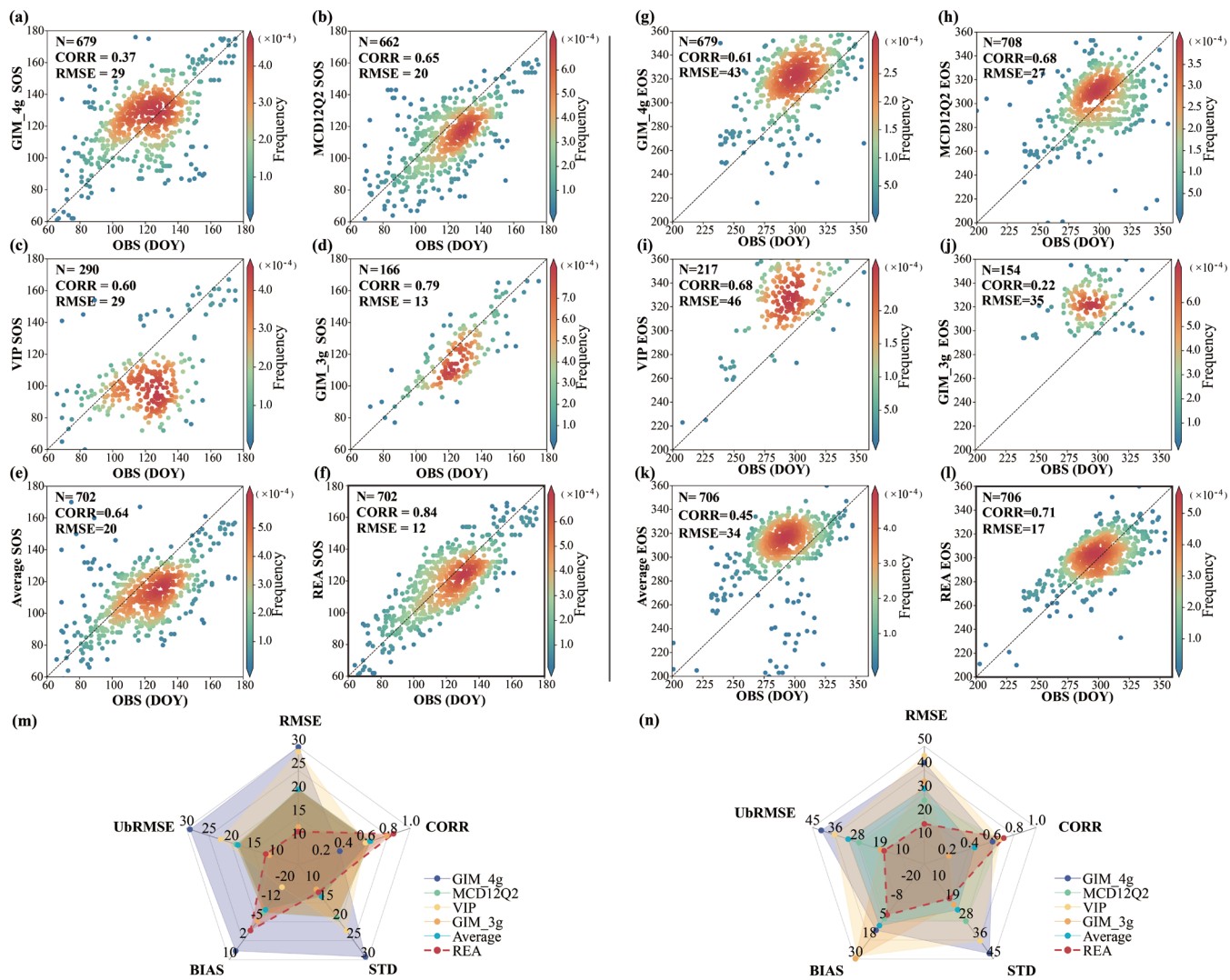

**Figure 5: Scatterplots and radar charts of performance for each phenology dataset and the merged phenology dataset obtained using the REA method. (**a–f) SOS evaluation results of the GIM_4g, MCD12Q2, VIP, GIM_3g, Average, and REA dataset, (m) radar chart of the SOS evaluation results, (g–l) EOS evaluation results of the GIM_4g, MCD12Q2, VIP, GIM_3g, Average, and REA dataset, and (n) radar chart of the EOS evaluation results. Each point represents a site year in the figure. OBS indicates ground-based phenocam phenological dates, RMSE indicates the root mean square error, UbRMSE indicates the unbiased RMSE, BIAS indicates the mean difference between the satellite-based results and the ground-based verification results, STD indicates the standard deviation, and CORR indicates the correlation coefficient. The radar chart is a graphical method used to display multivariate data in the form of a two-dimensional chart with axes starting from the same point.

We selected a long-term PhenoCam site (Morganmonroe) from the PhenoCam Network to evaluate the merged dataset across years at a single location. We have chosen an US PhenoCam site characterized by deciduous broad-leaved forest, with data from 2010–2020 for both SOS and EOS. As illustrated in the time series plot in Figure S5, the consistency between the REA results and PhenoCam data for both SOS and EOS is the highest when compared to the original four individual datasets.

Additionally, most vegetation phenology products demonstrate higher consistency with phenocam data for SOS compared to EOS.

### 3.4 Temporal trends of phenology based on the merged dataset

It is evident from Fig. 6(a) that the SOS exhibits a significant ($p < 0.01$) trend of advance (earlier dates in SOS) over the period 1982–2020, with a rate of advance of approximately 0.19 d yr$^{-1}$. Fig 6(b) presents the spatial distribution of the SOS trends obtained using the Mann–Kendall test. Approximately 64.37% of the regions exhibit a trend of advance, with 44.88% exhibiting a significant ($p < 0.05$) trend, while 18.53% of regions demonstrate a significant trend of delay (later dates in SOS).

Fig 6(c) illustrates that the EOS exhibits a significant trend of delay (later dates in EOS) with a rate of 0.18 d yr$^{-1}$ ($p < 0.01$). It is evident from Fig. 6(d) that the proportion of areas experiencing delayed EOS in regions above 30°N is 68.11% (comprising 46.60% significant at $p < 0.05$), consistent with the corresponding trend depicted in Fig. 6(c). Apart from the southwestern to northeastern regions of North America, Europe, the Middle East, and certain high-latitude areas in Asia, EOS delay is predominant.

Over the study area, 46.72% of regions exhibit SOS advance and EOS delay, 17.73% show SOS advance and EOS advance, 21.42% demonstrate SOS delay and EOS delay, and 14.14% show SOS delay and EOS advance.

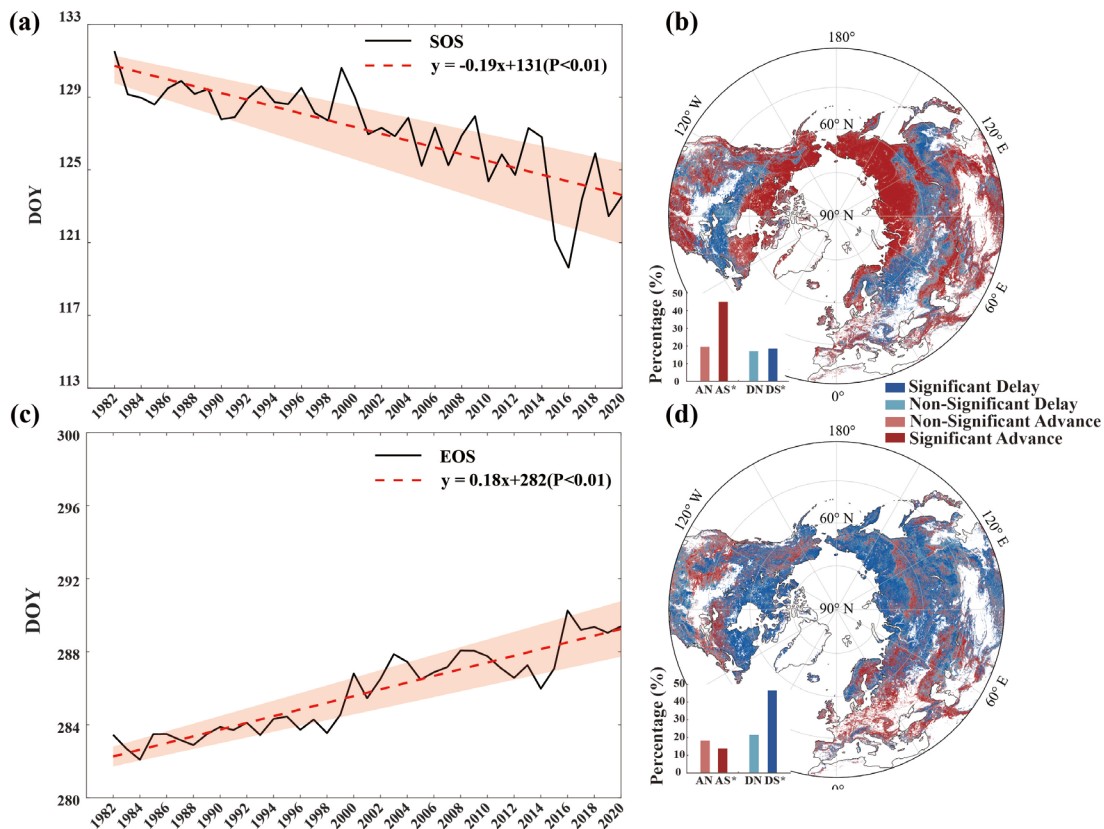

**Figure 6: Temporal and spatial trends of the SOS and the EOS over the period 1982–2020 based on the merged dataset obtained using the REA method.** (a) Temporal trend of the SOS over the period 1982–2020, (b) Spatial trend of the SOS over the period 1982–2020, (c)Temporal trend of the EOS over the period 1982–2020, (d) Spatial trend of the EOS over the period 1982–2020. The shaded area in (a) and (c) indicates uncertainty at one standard deviation, red lines in (a) and (c) are the trendlines of average SOS/EOS dates for each year, and black lines are the average SOS/EOS date for each year. The abbreviations DS (significant delay), DN (non-significant delay), AS (significant advance), and AN (non-significant advance) are used in the insets of panels b and d.

## 4 Discussion and Conclusions

### 4.1 Integrating Multi-Source Phenology Data and Addressing Dataset Discrepancies

This study integrates four widely used remote sensing phenology datasets (VIP, GIM_4g, MCD12Q2, and GIM_3g) using the Reliability Ensemble Averaging (REA) method to improve the consistency of SOS and EOS estimations. Our analysis shows substantial discrepancies among these datasets, consistent with previous findings that SOS estimates can vary by 60 days (White et al., 2009) or even exceed 100 days (De Beurs and Henebry, 2010), depending on the extraction method. We find the inconsistencies that arise from both differences in data sources and variations in phenology extraction methodologies by evaluating the spatial distribution of the multi-year mean SOS and EOS across datasets.

The REA method produces a more harmonized phenology result by assigning different weights to each dataset based on its reliability to mitigate discrepancies. The effectiveness of this approach is demonstrated by the lower deviation of SOS and EOS estimates between merged results and phenocam datasets. As shown in Figure 1, the differences in SOS and EOS among the four datasets across the Northern Hemisphere reach approximately 15 days on average, with some regions exhibiting even larger discrepancies (more than 120 days). These variations might be caused by factors such as sensor characteristics, data preprocessing techniques, and phenological extraction methods. For instance, similarities in spatial distributions between the GIM_4g and VIP datasets likely stem from their common use of AVHRR sensor data, whereas their mean difference (14 days) could be attributed to differences in time-series reconstruction techniques. Differences between GIM_4g and GIM_3g are primarily driven by different data fitting methodologies. The data smoothing techniques aim to preserve vegetation dynamics while reducing noise, but the optimal method is still unclear, as the appropriate method depends on the specific biogeographical characteristics of a given region (Hird and McDermid, 2009).

Additionally, phenology extraction methods exhibit varying effectiveness across different ecological regions (Reed, 2006). Evaluating dataset consistency with PhenoCam observations across different plant functional types (PFTs), we find that dataset performance varies by PFT and time period. For instance (in Figure S2), VIP data show consistency with PhenoCam observations in deciduous broadleaf forests in 2001, whereas GIM_3g outperformed other datasets in 2002. And MCD12Q2 shows better agreement in evergreen needleleaf forests. These findings stress the challenges of integrating datasets with inherently different methodological foundations.

We further evaluated the greening trend, which reflects the temporal change in average vegetation greenness during the growing season, as indicated by NDVI trends (Li et al., 2023). The greening trend estimated from VIP ($-10.16 \times 10^{-4}$ yr$^{-1}$) was lower than those from REA ($-1.14 \times 10^{-4}$ yr$^{-1}$) and GIM_4g ($3.55 \times 10^{-4}$ yr$^{-1}$). The proportion of areas showing greening trends was 25.22% (with 41.44% significantly greening) for VIP, 68.49% (with 38.32% significantly greening) for GIM_4g, and 49.83% (with 56.83% significantly greening) for REA. These substantial differences in greenness trends across datasets (Figure S3) highlight the necessity of an integrated approach for robust phenology estimation.

**4.2 Strengths, Limitations, and Sensitivity Analysis of the REA Approach**

Given the lack of consensus on the "best" approach for extracting phenological dates, our study highlights the value of integrating multiple methodologies rather than relying on a single dataset. The REA method improves phenology estimations by weighting datasets based on reliability rather than assuming equal contributions, as in a simple arithmetic average (Lu et al., 2021). The REA method considers the temporal correlation of vegetation phenology data by employing a voting principle (Giorgi and Mearns, 2002), and this approach facilitates convergence of data while retaining differences of the spatial distribution, thereby offering advantages in multisource data fusion than simple averaging. Our results indicate that REA merged phenology results are more consistent with PhenoCam observations than individual datasets and simple average results,

demonstrating its potential to reduce biases between different datasets caused by different extraction methods. However, while REA enhances consistency, it does not fully eliminate methodological discrepancies.

To evaluate the robustness of this approach, we conducted sensitivity analyses using different dataset and time span combinations. Our results (see Figures S6 and S7) indicate that the combination of different input datasets has a greater influence on fusion results than the length of the time series. The average deviation of SOS/EOS dates between different dataset combinations and the long-term fusion result using all four datasets is 7.1 days, whereas SOS and EOS variations across different fusion time spans (e.g., 5 year vs. 10-year fusion) are relatively minor (4.1 and 3.2 days, see Figure S6). Moreover,

fusion weights derived from shorter time series are largely consistent with those obtained using the full dataset (see Figure S8). In the sensitivity analysis assessing the impact of fusion time span, the SOS results were more stable than EOS, and fusion results based on longer time series were more similar to the full-period REA fusion result (1982–2020).

    The effectiveness of data fusion depends on the complementarity and quality of input datasets. Our analysis suggests that datasets incorporating more reliable sources could produce improved fusion results. For example, GIM_3g SOS demonstrate

higher consistency with phenocam observations (Figure 5d); groups 2 and 3, which include GIM_3g SOS, exhibit smaller deviations from the long-term REA fusion results than group 1. In contrast, EOS estimates are more stable across different dataset combinations. These results show that multi-source data integration could help improve accuracy. The similarity in weight distributions across different dataset combinations (Figure S9) suggests that dataset selection remains a critical factor influencing final estimates. Incorporating additional high-quality datasets in future studies could further enhance accuracy and

robustness.

### 4.3 Consistency of SOS and EOS Trends Across Studies and Key Influencing Factors

    Global climate change has significantly influenced vegetation phenology, with a general advancement in SOS and a delay in EOS (Piao et al., 2019). Our REA-based dataset estimates an SOS advancement of 0.19 days per year during 1982–2020 ($p < 0.01$), aligning with previous findings, such as $1.4 \pm 0.6$ days per decade for 1982–2011 (Wang et al., 2015) and a 5.4-

490 day advance from 1982 to 2008 (Jeong et al., 2011). Similarly, EOS has been delayed at a rate of 0.18 days per year ($p < 0.01$), consistent with prior estimates of $0.18 \pm 0.38$ days per year for 1982–2011 (Liu et al., 2016) and a 6.6-day delay from 1982 to 2008 (Jeong et al., 2011). Deviations between REA-based trends and those from individual datasets (Blunden et al., 2023) suggest that variations in data sources and study regions contribute to observed discrepancies.

    Differences in SOS trends across studies arise from multiple factors, including study period, land cover changes, spatial

domain, environmental heterogeneity, and methodological differences. Trend estimates are highly sensitive to the selected time window, particularly at the start and end years (Cong et al., 2013). Interannual and decadal climate variability can further influence these trends contributing to discrepancies across studies. Land cover changes, such as wildfires and deforestation (Jeong et al., 2011), can influence the extraction result of phenology, then affect SOS trends. Additionally, spatial differences

in studies also influence the rate of trends in different studies, as phenology is affected by vegetation type and climate sensitivity. For instance, Jeong et al. (2011) analyzed temperate vegetation between 30°N – 80°N (SOS advance in 0.2 days per year), and Wang et al. (2015) focused on 30°N – 75°N (SOS advance in 0.14 $\pm$ 0.6 days per year), excluding evergreen forests and managed landscapes. Since phenology responds differently across species and locations (Maignan et al., 2008), variations in study areas contribute to discrepancies. Photoperiod constraints introduce additional latitudinal differences in phenological responses to warming (Meng et al., 2021). Moreover, precipitation and other environmental factors vary spatially, further influencing SOS estimates.

Different methods of extracting phenology from remote sensing data could introduce uncertainties between different datasets (Cong et al., 2013; Jeong et al., 2011; White et al., 2009). Differences in filling missing data and filtering methods can affect the continuity of the vegetation index time series, leading to discrepancies in phenology results. Reducing these inconsistencies among different datasets and producing more harmonized data is a primary motivation for employing the REA method, which reduces uncertainties by integrating datasets based on their reliability. Recognizing these factors also explains the discrepancies in SOS/EOS trends and stresses the need for standardized methodologies in phenological trend analyses.

**4.4 Conclusion and Perspective**

Shifts in vegetation phenology affect ecosystem structure (Kharouba et al., 2018; Yang and Rudolf, 2010), consequentially affecting biodiversity (Renner and Zohner, 2018), terrestrial carbon and water cycles (Piao et al., 2020), and the climate system (Green et al., 2017; Piao et al., 2020). The establishment of a comprehensive and reliable vegetation phenology dataset is thus of critical importance. Our study demonstrated that the REA method provides a robust approach for integrating multi-source phenology datasets and produced a vegetation phenology dataset for regions above 30 degrees in the Northern Hemisphere from 1982 to 2020. This dataset could be used for subsequent analyses, such as examining vegetation phenology dynamics and their impacts on the terrestrial carbon cycle and water balance and providing climatic feedback for global vegetation dynamics modeling. Integrating more vegetation phenology datasets that consider regional characteristics and refining the weight could be done in a future study, which could improve the accuracy and reliability of the merged phenology dataset. Additionally, higher-resolution phenology datasets will also improve the consistency between remote sensing phenology datasets and ground-based results. High-resolution datasets will enhance our ability to assess phenological changes in heterogeneous landscapes and improve local-scale ecological modeling, offering new opportunities to enhance monitoring and prediction of vegetation dynamics in response to environmental change.

**Data availability**

The MCD12Q2 phenology dataset is available at https://lpdaac.usgs.gov/products/mcd12q2v061/(Friedl et al., 2022), the VIP phenology dataset is available at https://lpdaac.usgs.gov/products/vipphen_ndviv004/ (Didan and Barreto, 2016), the GIM_3g

phenology dataset is available at http://data.globalecology.unh.edu/data/GIMMS_NDVI3g_Phenology/ (Wang et al., 2019),
the GIM_4g phenology dataset is available at https://doi.org/10.5281/zenodo.11136967 (Chen and Fu, 2024), the camera-based
phenology dataset is available at https://daac.ornl.gov/, http://www.sizenken.biodic.go.jp/, and http://www.pheno-eye.org/, the
land use dataset is available at https://lpdaac.usgs.gov/products/mcd12q1v061/ (Friedl and Sulla-Menashe, 2022), and the REA
phenology dataset is available at https://doi.org/10.5281/zenodo.15165681 (Cui and Fu, 2024).

## Competing interests

The contact author declares that none of the authors has any competing interests.

## Author contribution

YHF developed the preliminary conceptualization; YC performed the study, YC, YHF and YZ wrote the manuscript; YHF and
YZ supervised the manuscript construction and revision; SC, YG, ML, ZJ participated in reviewing and editing the paper. All
authors have read and approved the paper.

## Acknowledgments

The work was supported by the National Funds for Distinguished Young Youths (Grant No. 42025101), the National Key
Research and Development Program of China (Grant No. 2023YFF0805604), the Fundamental Research Funds for the Central
Universities (2243300004), and the 111 Project (Grant No. B18006). We are grateful to the anonymous reviewers and editor
for their valuable suggestions that helped us strengthen the quality of our paper. We thank Liwen Bianji for editing the English
text of a draft of this manuscript. Data used in this research were provided by the PhenoCam Network, which has been
supported by the National Science Foundation, the Long-Term Agroecosystem Research (LTAR) network which is supported
by the United States Department of Agriculture (USDA), the U.S. Department of Energy, the U.S. Geological Survey, the
Northeastern States Research Cooperative, and the USA National Phenology Network. We thank the PhenoCam Network
collaborators, including site PIs and technicians, for publicly sharing the data that were used in this paper.

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
