# Peer review of "A vegetation phenology dataset by integrating multiple sources using the Reliability Ensemble Averaging method"

_Earth System Science Data, 2024_

## Author Comment (AC1)

**Response to Reviewer #2:**

[Comment 1] The paper integrates four existing phenology datasets by applying different weights to the start (SOS) and end (EOS) of the growing season, as determined by each dataset. Overall, the manuscript is well-written and the dataset presented would be useful for community. However, several issues need to be addressed before it can be accepted for publication.

Response: We thank the reviewer for the supportive and constructive comments, and we appreciate the reviewer's assistance in the acknowledgments section in the revised manuscript. We have addressed each point and adjusted the manuscript according to the reviewer's comments. Please find below a detailed response to each comment.

**Major concerns:**

[Comment 2] First, the remote sensing datasets are not processed exactly the same, in terms of curves used to fit the time series and threshold used to extract transition dates. the authors should clarify how these methodological differences might influence the uncertainty or accuracy of the resulting merged dataset.

Response: Thank you for your comment. Yes, indeed different methods were applied in different dataset. In MCD12Q2 dataset, the time series data was fitted by a penalized cubic smoothing spline, and Greenup (dormancy) is defined as the date when the EVI2 time series first (last) crosses 15% of the segment EVI2 amplitude. In VIP dataset the filtering method based on confidence interval and operational continuity algorithm were used to rebuild the time series curves, the start (end) of season is defined using the modified Half-Max method as the date when the NDVI2 time series first (last) crosses 35% of the segment NDVI2 amplitude. In GIM_3g dataset, a double logistic function was applied to fit the NDVI curve, and it uses the date when the NDVI first (last) crosses 20% of the segment NDVI amplitude as the SOS (EOS). In GIM_4g dataset, the NDVI time series data were fitted and smoothed using five fitting methods: the HANTS-Maximum, Spline-Midpoint, Gaussian-Midpoint, Timesat-SG, and Polyfit-Maximum methods, and it uses the date when the NDVI first (last) crosses 20% (50%) of the segment NDVI amplitude as the SOS (EOS). The smoothing method and phenology extraction method differs in these datasets.

Among different methods for vegetation phenology extraction, it is hard to distinct the best method for extracting vegetation phenology. According to previous and the present study, the phenology estimates obtained from different extraction methods show significant variation, with the estimated results differing by up to one month or more than 60 days depending on the method applied across different regions (Cong et al., 2012).

The phenological dates that were extracted from different methods were supposed to indicate changes in actual physiological conditions as accurately as possible, and the average method is often used for the fusion of different datasets, however, the effectiveness of these methods varies across regions and time period even, and may not always represent the true vegetation conditions. Different vegetation phenology datasets show different performance across regions and years comparing to the phenocam dates in the Fig. S1 below, for example, the consistency of VIP and ground phenocam data in the year 2001 of deciduous broadleaf forest is the best, whereas only

the consistency of GIM_3g data in the year 2002 is better than that of VIP data. Comparing to the forest types, the consistency of remote sensing based phenological dates and phenocam data is higher in deciduous broadleaf region when using GIM_3g method, but in evergreen needleleaf when using the MCD method. We want to get the results that best reflect the physiological state at different sites and years.

Therefore, we use the REA method to catch the dates which can best reflects the change of the vegetation growing state based on the assumption that there exists a data source capable of reflecting the vegetation conditions at each gridcell, and different weights assigned to each data are calculated based on their reliability to get the final result.

To avoid confusion and clarify this issue, we revised the corresponding text, please refer to line 376-385 and 398-404 in the revised manuscript.

Cong, N., Piao, S., Chen, A., Wang, X., Lin, X., Chen, S., Han, S., Zhou, G., and Zhang, X.: Spring vegetation green-up date in China inferred from SPOT NDVI data: A multiple model analysis, *Agricultural and Forest Meteorology*, 165, 104 – 113, https://doi.org/10.1016/j.agrformet.2012.06.009, 2012.

Wu, W., Sun, Y., Xiao, K., and Xin, Q.: Development of a global annual land surface phenology dataset for 1982–2018 from the AVHRR data by implementing multiple phenology retrieving methods, *International Journal of Applied Earth Observation and Geoinformation*, 103, 102487, https://doi.org/10.1016/j.jag.2021.102487, 2021.

Zeng, L., Wardlow, B. D., Xiang, D., Hu, S., and Li, D.: A review of vegetation phenological metrics extraction using time-series, multispectral satellite data, *Remote Sensing of Environment*, 237, 111511, https://doi.org/10.1016/j.rse.2019.111511, 2020.

Zhang, J., Zhao, J., Wang, Y., Zhang, H., Zhang, Z., and Guo, X.: Comparison of land surface phenology in the Northern Hemisphere based on AVHRR GIMMS3g and MODIS datasets, *ISPRS Journal of Photogrammetry and Remote Sensing*, 169, 1–16, 2020.

[Figure]

**Figure S1: SOS RMSE of four datasets in different PFT for the period 2001–2018.** Four datasets refer to GIM_4g, MCD12Q2, VIP, and GIM_3g datasets, respectively. PFT: plant functional type, DB: deciduous broadleaf, DN: deciduous needleleaf, EN: evergreen needleleaf, GR: grassland, SH: shrubs, TN: tundra, WT: wetland. The black boxes represent the best data for the year in that PFT.

**[Comment 3]** Second, I question whether the REA method truly outperforms a simple average. I recommend that the authors include additional analysis comparing the results obtained using the REA method with those from a simple average.

**Response:** Thank you for your thoughtful comment. Following the reviewer's suggestion, we have added the comparison between simple average and REA method result, see below Fig. 5(e) & (k). Comparing with simple average, the REA-based SOS shows better performance in RMSE (REA and Average, 12d and 21d, respectively), CORR (REA and Average, 0.84 and 0.65, respectively), BIAS (REA and Average, -1.5d and -9.7d, respectively) and UbRMSE (REA and Average, 12d and 18d, respectively). The REA-based EOS also shows better performance in RMSE (REA and Average, 17d and 32d, respectively), CORR (REA and Average, 0.71 and 0.45, respectively), BIAS (REA and Average, 1.0d and 8.0d, respectively) and UbRMSE (REA and Average, 17d and 31d, respectively). We also calculated the mean SOS and EOS using simple average for the period 1982-2020 in Fig. S3, there was little difference in the overall spatial distribution patterns of simple average and REA results, but the specific dates differ. We revised the figure and added the new results in the revised manuscript, please refer

[Figure]

**Figure 5: Scatterplots and radar charts of performance for each phenology dataset and the merged phenology dataset obtained using the REA method. (**a–f) SOS evaluation results of the GIM_4g, MCD12Q2, VIP, GIM_3g, Average, and REA datasets, respectively, (m) radar chart of the SOS evaluation results, (g–l) EOS evaluation results of the GIM_4g, MCD12Q2, VIP, GIM_3g, Average, and REA datasets, respectively, and (n) radar chart of the EOS evaluation results. Each point represents a site year in the figure. OBS indicates ground-based phenocam phenological dates, RMSE indicates the root mean square error, UbRMSE indicates the unbiased RMSE, BIAS indicates the mean difference between the satellite-based results and the ground-based verification results, STD

[Figure]

indicates the standard deviation, and CORR indicates the correlation coefficient.

**Figure S3: Mean (a) SOS and (b) EOS dates (DOY) obtained using simple average for the period 1982 – 2020.**
RMSE indicates the root mean square error (day), CORR indicates the correlation coefficient.

**[Comment 4]** PhenoCam data are not properly cited; please check out the fair use data policy here https://phenocam.nau.edu/webcam/fairuse_statement/.

**Response:** We thank the reviewer for point out this mistake. We have correctly cited the PhenoCam data both in the text and the acknowledgement in the manuscript.

**Line edits:**

**[Comment 5]:** PhenoCam, as a ground-based measurement, has been operational for more than 20 years. It should be introduced earlier in the text here.

**Response: Following the reviewer's suggestion, w**e have added the introduction of PhenoCam earlier in the text. PhenoCam, as a ground-based measurement, has been operational for more than 20 years (Richardson et al., 2018a). Please refer to line 31 in the revised manuscript.

**[Comment 6]:** What is the specific time period over are evaluated?

**Response:** The trends of vegetation phenology from GIMMS3g and MODIS were evaluated during 2000-2015. This information has been added in the text, please refer to line 44-45 in the revised manuscript.

**[Comment 7]:** Please provide examples of regions where significant differences in the phenological metrics are observed.

**Response:** Please see the response to comment#2 for the same question.

**[Comment 8]:** As you mentioned earlier, the performance of datasets varies across regions. How does the REA method address or resolve these regional performance variations?

**Response:** Thank you for your comment. The REA method merged different datasets for a better performance globally, we assume that significant deviations are unlikely to occur simultaneously across most data sources within a specific region. Therefore, data containing anomalies will be excluded to achieve more accurate results using the REA method. If there is one data that shows significant discrepancies compared to other data, which may cause by improper extraction methods in that region, the $B_{Phe,i}$ and $D_{Phe,i}$ will extract this variance and combine with the natural variability $\varepsilon_{Phe}$ of the region in the weight distribution process. If the natural variability of that region is low, a smaller value is assigned to the weight, and if the natural variability of the region is large, the weight is assigned by both the natural variability and the deviations. This is why the REA method demonstrates robust performance across all regions. To clarify this issue, we updated the corresponding text, please refer to line 201-204 in the revised manuscript.

**[Comment 9]:** How are SOS and EOS determined in the VIP phenology dataset? Please compare these criteria with the methods used to determine greenup in the MCD12Q2 dataset.

**Response:** Following the reviewer's suggestion, we added details information about criteria in these methods. In details, the start (end) of season is defined using the modified Half-Max method as the date when the NDVI2 time series first (last) crosses 35% of the segment NDVI2 amplitude in VIP. Greenup (dormancy) is defined as the date when the EVI2 time series first (last) crosses 15% of the segment EVI2 amplitude

in MCD12Q2. Please refer to line 103-107 and 94-95 in the revised manuscript.

**[Comment 10]**: What specific curves are applied for the MCD12Q2 and VIP datasets?
**Response:** The time series data was fitted by a penalized cubic smoothing spline to rebuild time series curve in MCD12Q2 dataset (line 90). The filtering method based on confidence interval and operational continuity algorithm were used to rebuild the time series curves in VIP dataset. We revised the corresponding text, please refer to line 95 and line 103-104 in the revised manuscript.

**[Comment 11]**: What threshold is used to extract phenological metrics from the GIM_3g dataset?
**Response:** This product provides phenology data for the Northern Hemisphere, and it uses the date when the NDVI first (last) crosses 20% of the segment NDVI amplitude as the SOS (EOS). We revised the corresponding text, please refer to line 112 in the revised manuscript.

**[Comment 11]**: Please provide a link to the PhenoCam dataset for reference.
**Response:** We have added the website of PhenoCam_V2 (https://daac.ornl.gov/VEGETATION/guides/PhenoCam_V2.html) and the PhenoCam (https://phenocam.nau.edu/webcam/) in the text. Please refer to line 133 in the revised manuscript.

**[Comment 12]**: Since the method relies on interannual variability in the time series, what is the minimum required length for the time series? Is it possible to use REA to merge only two datasets? A discussion or quick test with shorter time series would be valuable, especially considering the availability of recent Planet data with.
**Response:** Thank you for your comment and following the reviewer's suggestion, we updated the discussion section. In the method proposed by Giorgi et al. 2001, there is no restriction on the minimum value for this parameter. It can be adjusted multiple times according to the actual data to find an appropriate range for the specific dataset, but it should be as large as possible to reflect the natural variability. It is possible to get the result by merging two datasets with REA, but the accuracy may be less than that of merging with more reliable data sources. Please refer to the revised text in line 415-416 in the revised manuscript.

Giorgi, F. and Mearns, L. O.: Calculation of Average, Uncertainty Range, and Reliability of Regional Climate Changes from AOGCM Simulations via the "Reliability Ensemble Averaging" (REA) Method, *Journal of Climate*, 15, 1141–1158, https://doi.org/10.1175/1520-0442(2002)0152.0.CO;2, 2002.

**[Comment 13]**: For datasets with higher interannual variability, did you assign them lower weights in the REA method? Please clarify.
**Response:** Interannual variability is measured by $\varepsilon_{Phe}$ in equation (2), which is also represents for natural variability. Natural variability changes from region to region, in Equation (1) and (6), $\varepsilon_{Phe}$ cancels out under the condition of $B_{Phe,i}$ and $D_{Phe,i}$ greater than $\varepsilon_{Phe}$, which based on the assumption that more stringent on are required to increase the reliability over regions characterized by lower natural variability. The natural

variability does not work single, it works with $B_{Phe,i}$ and $D_{Phe,i}$ jointly. For the region in lower natural variability, if the phenology data from one dataset also have large difference with other datasets, it is given lower weight for generate the REA phenology at that region, which is thought to be less accurate data at that region. To avoid confusion, we revised the corresponding text, please refer to line 189-194 and 201-204 in the revised manuscript.

**[Comment 14]**: Open-source code for the REA method should be made available. This would assist the community in merging datasets from various sources and years.
**Response:** The code is shared on Github (https://github.com/PRqA642/REA).

**[Comment 15]**: Please provide a brief description of the metrics used and their characteristics.
**Response:** We have supplemented the description to our manuscript. The RMSE is calculated as the square root of the average of the squares of the residuals, which penalizes larger errors than smaller ones and provide an estimate of the magnitude of errors between remote sensing estimated value and phenocam datasets. BIAS is the average difference between remote sensing estimated value and phenocam value, that helps in understanding whether the estimated value is higher or lower than phenocam value. The correlation coefficient measures the linear relationship between two variables. The ubRMSE measures the deviation between two variables without systematic errors. Standard deviation quantifies the variation of the dataset, which measures the deviation between data and the mean value. Please refer to line 217-223 in the revised manuscript.

**[Comment 16]**: Provide a specific example of how the M-K test will be applied in the study.
**Response:** We used M-K test to analyze the trend of SOS and EOS during 1982-2020 in the merged dataset, and we have supplemented this part of content. We have added how we will use the M-K test in the method part. Please refer to line 227-228 in the revised manuscript.

**[Comment 17]**: The citation of Mao and Sun may not be necessary here—consider removing it.
**Response:** Following the reviewer's suggestion, we have removed the citation of Mao and Sun from the text in the revised manuscript.

**[Comment 18]**: I am a bit concerned about the huge deviations in spring 2022 shown by Figure 7, which seems very inconsistent with Figure 2.78 in DOI: https://doi.org/10.1175/BAMS-D-23-0090.1
**Response:** Thank you for your comment. During 2022, there is only MCD12Q2 left as the data source, since other dataset do not include this time period and large uncertainty may exist, so we remove the data after 2020. In the revised manuscript, we estimated the trends in SOS and EOS until 2020, please see the revised figure 6.

[Figure]

**Figure 6: Temporal and spatial trends of the SOS and the EOS over the period 1982–2020 based on the merged dataset obtained using the REA method.** (a) Temporal trend of the SOS over the period 1982-2020, (b) Spatial trend of the SOS over the period 1982-2020, (c)Temporal trend of the EOS over the period 1982-2020, (d) Spatial trend of the EOS over the period 1982-2020. The shaded area in (a) and (c) indicates uncertainty at one standard deviation, red lines in (a) and (c) are the fitting lines of average SOS/EOS dates for each year, and black lines are the average SOS/EOS date for each year. Significant delay (DS), non-significant delay (DN), significant advance (AS), non-significant advance (AN).

**[Comment 19]:** A description of how uncertainty is determined needs to be added to the REA phenology dataset.

**Response:** The calculation of uncertainty is introduced in the method part, please refer to line 195-200 in the revised manuscript. The uncertainty range is calculated based on the weight of each dataset and the deviation between REA result and data sources, the upper and lower uncertainty limits are measured by REA result and the uncertainty range.

---

## Author Comment (AC2)

**Response to Reviewer #1:**

**[Comment 1]** This data paper describes a new vegetation phenology dataset that fuses four existing remotely sensed datasets. It then compares the timing of two phenophases (start and end of the growing season) from this new dataset to the same metrics from three phenology camera networks (phenocams). Considering the discrepancies in accuracy and temporal/spatial coverage between existing phenology datasets, this new fusion dataset, created based on weighed averaging, is extremely useful and appears to have higher accuracy than existing datasets. However, the methods could be explained in greater detail, in addition to a few other concerns, which I highlight below.

**Response:** We are grateful to the reviewer for recognizing the extremely useful and importance of our research and for the constructive comments. We also thank the reviewer for these thoughtful and constructive comments and suggestions, which have substantially improved our manuscript. We appreciate the reviewer's assistance in the acknowledgments section of the revised manuscript. We have addressed each point and adjusted the manuscript according to the reviewer's comments. Please find below a detailed response to each comment.

**Major concerns:**

**[Comment 2]** First, I'm concerned about how the SOS and EOS dates were extracted and compared across the different datasets. Of the 4 datasets fused together, at least several appear to use different SOS and EOS thresholds and methods to extract the dates, which begs the questions – are they directly comparable? For example, if one dataset identifies SOS as 15% of maximum green-up, but another uses 50%, the extracted SOS date will naturally differ between those datasets, but one is not necessarily more or less accurate than the other. The same applies when comparing the fused dataset with the phenocam dataset – how are SOS and EOS identified across the 3 phenocam networks included in the phenocam dataset? Also, in addition to different thresholds, it appears that the datasets use different methods to extract SOS and EOS dates. This could be one reason that there is observed variability across the SOS and EOS dates. If possible, it would be best to standardize the methods and threshold across all the datasets. It might also be helpful to compare entire seasonal trends in vegetation greenness through time to visualize differences between datasets instead of just the seasonal transition dates.

**Response:** We thank the reviewer for his/her thoughtful comment. We totally agree that different thresholds were used in different extraction methods and may induct large variation in phenological dates, as previous studies reported that there are differences in vegetation phenology results obtained from various remote sensing phenology algorithms (time series data processing methods and phenology extraction methods) (Cong et al, 2012; Wu et al, 2021; Zeng, 2020).

Remote sensing vegetation phenology typically reflects transition dates in the vegetation growth cycle, such as the start (leaf out) and end (leaf senescence) of the growing season, and different vegetation indices, e.g. NDVI, LAI and SIF, were used. The phenological dates that were extracted from different methods were supposed to

indicate changes in actual physiological conditions as accurately as possible. However, the effectiveness of these methods varies across regions and time period, and may not always represent the true vegetation conditions. Such as, different vegetation phenology datasets show different performance across regions and years comparing to the phenocam dates, see the Fig. S1 below. The consistency of VIP and ground phenocam data in the year 2001 of deciduous broadleaf forest is the best, whereas only the consistency of GIM_3g data in the year 2002 is better than that of VIP data. Comparing to the forest types, the consistency of remote sensing based phenological dates and phenocam data is higher in deciduous broadleaf region when using GIM_3g method, but in evergreen needleleaf when using the MCD method. We want to get the results that best reflect the physiological state at different sites and years. Therefore, a method that integrates data from different methods based on reliability to combine the advantages of different data sources is feasible. In the revised manuscript, we clarify this argument, please refer to line 375-386 in the revised manuscript.

In PhenoCam, Spline interpolation method was applied to extract transition dates for each ROI mask in PhenoCam Dataset v2.0. We used the date when the GCC first (last) crosses 25% of the GCC amplitude as the SOS and EOS. The second phenocam dataset is from the Japan Internet Nature Information System digital camera data (http://www.sizenken.biodic.go.jp/) acquired over the period $2002-2009$. Ide and Oguma (Ide and Oguma, 2010) provided greenup dates for two phenocam sites with areas of interest (AOI) defined at the species level scale. The vegetation types included in their data comprise wetland and mixed deciduous forest. The date of green-up each year was estimated as the DOY of the maximum rate of increasing 2G-RBi (i.e., the maximum of the second derivative. The third dataset consists of phenology data for deciduous broadleaf forests in Japan (Inoue et al., 2014), supported by the Phenological Eyes Network (http://www.pheno-eye.org/), which is a network of ground-based observatories for long-term automatic observation of vegetation dynamics established in 2003 (Nasahara and Nagai, 2015), the start and end of season is defined as the first day when 20% of leaves had flushed and the first day when 80% of leaves had fallen in the given ROI, respectively.

We thank the reviewer for recommending to standardize the methods and threshold across all the datasets, whereas it is difficult to standardize the various methods and thresholds, because standardizing thresholds requires the acquisition of a complete vegetation index time series. This approach may diminish the significance of data fusion, i.e. the selection of different thresholds and smoothing methods for the data source that best reflect the changes in the physiological state of the vegetation.

Following the reviewer's suggestion, we estimated the entire seasonal trends in vegetation greenness, e.g. the NDVI trends, for VIP, GIM_4g and REA method, please see the results in Fig. S2 below. The average greening trend in VIP $(-10.16 \times 10^{-4}/yr)$ is lower than REA $(-1.14 \times 10^{-4}/yr)$ and GIMMS $(3.55 \times 10^{-4}/yr)$. The greening rate in VIP, GIM_4g and REA are 25.22% (with 41.44% significantly greening), 68.49% (with 38.32% significant greening) and 49.83% (with 56.83% significant greening), respectively. Large difference in these greenness trends were found, supporting our findings and an integrated method is thus needed to produce a dataset that combing the advantages of different data sources. In the revised manuscript,

we have added this argument in the discussion section, please refer to line 437-441.

**Figure S1: SOS RMSE of four datasets in different PFT for the period 2001 − 2018.** Four datasets refer to GIM_4g, MCD12Q2, VIP, and GIM_3g datasets, respectively. PFT: plant functional type, DB: deciduous broadleaf,

[Figure]

DN: deciduous needleleaf, EN: evergreen needleleaf, GR: grassland, SH: shrubs, TN: tundra, WT: wetland. The black boxes represent the best data for the year in that PFT.

[Figure]

**Figure S2: Growing season NDVI trend obtained using (a)VIP phenology dataset, (b) GIM_4g dataset and (c) REA phenology dataset for the period 1982－2015 and (d) the distribution of their frequency.** GS: growing season. Black dots represent for regions where the trend is significant (P<0.05).

Cong, N., Piao, S., Chen, A., Wang, X., Lin, X., Chen, S., Han, S., Zhou, G., and Zhang, X.: Spring vegetation green-up date in China inferred from SPOT NDVI data: A multiple model analysis, *Agricultural and Forest Meteorology*, 165, 104－113, https://doi.org/10.1016/j.agrformet.2012.06.009, 2012.

Wu, W., Sun, Y., Xiao, K., and Xin, Q.: Development of a global annual land surface phenology dataset for 1982–2018 from the AVHRR data by implementing multiple phenology retrieving methods, *International Journal of Applied Earth Observation and Geoinformation*, 103, 102487, https://doi.org/10.1016/j.jag.2021.102487, 2021.

Zeng, L., Wardlow, B. D., Xiang, D., Hu, S., and Li, D.: A review of vegetation phenological metrics extraction using time-series, multispectral satellite data, *Remote Sensing of Environment*, 237, 111511, https://doi.org/10.1016/j.rse.2019.111511, 2020.

**[Comment 3]** Also, this is mostly semantics, but an important distinction to not mislead readers. "PhenoCam" with a capital P and C is usually used to indicate data from the PhenoCam Network (phenocam.nau.edu) and phenocam (lowercase p and c) indicates a generic camera from any network. Please see Richardson 2023, Box 3 for further explanation (https://www.sciencedirect.com/science/article/pii/S0168192323004410). Since you used digital camera data from 3 different sources, please use "phenocam" in this paper to avoid confusion.

**Response:** We thank the reviewer for pointing out this mistake, and have corrected the use of "PhenoCam" and "phenocam" in the revised manuscript.

**[Comment 4]** Finally, I know discussion sections are normally shorter in data papers, but it seems like the authors could connect their results more to other studies. For example, how does the rate of growing season advance/delay compare to the rate that other studies have found? The fact that this analysis is done makes this more than just a data paper, and requires more contextualization with the literature. I would suggest either removing that analyses or adding more sources to support your results.

**Response:** Following the reviewer's suggestion, we have added more details in the discussion about the rate of phenological changes.

Both advanced spring phenology and delayed autumn phenology were found between REA-based phenological dates and previous studies. But the amplitudes of trends are different among these studies. In details, SOS was found significantly advance at the rate of 0.19 days per year in the REA result during 1982-2020, while the advancing rate of $1.4 \pm 0.6$ days per decade during 1982-2011 (Wang et al., 2015) and 5.4 days advanced from 1982 to 2008 (Jeong et al., 2011) was also found in previous studies. Similarly, EOS was found significantly delayed at the rate of 0.18 days per year in the REA result over the same period, while the $0.18 \pm 0.38$ days per year delay was found for 1982-2011 (Liu et al., 2016) and the 6.6 days delay was found from 1982 to 2008 (Jeong et al., 2011) in previous studies. In the revised manuscript, we added these statements in the discussion section in line 428-435.

Piao, S., Liu, Q., Chen, A., Janssens, I. A., Fu, Y., Dai, J., Liu, L., Lian, X. U., Shen, M., and Zhu, X.: Plant phenology and global climate change: Current progresses and challenges, *Global change biology*, 25, 1922–1940, https://doi.org/10.1111/gcb.14619, 2019.

Wang, X., Piao, S., Xu, X., Ciais, P., MacBean, N., Myneni, R. B., and Li, L.: Has the advancing onset of spring vegetation green‐up slowed down or changed abruptly over the last three decades?, *Global Ecology and Biogeography*, 24, 621 – 631, https://doi.org/10.1111/geb.12289, 2015.

Jeong, S.-J., HO, C.-H., GIM, H.-J., and Brown, M. E.: Phenology shifts at start vs. end of growing season in temperate vegetation over the Northern Hemisphere for the period 1982–2008, *Global change biology*, 17, 2385–2399, https://doi.org/10.1111/j.1365-2486.2011.02397.x, 2011.

**Line edits:**

**[Comment 5]**: Here and elsewhere, I would suggest saying "a ground-based phenocam data" instead of "the ground-based…"- using "the" makes it sounds like you're

referring to a single data source, (such at the PhenoCam Network), but your phenocam dataset actually includes multiple data sources. Also, see my comment for line 109 about using "PhenoCam" versus "phenocam"

**Response:** Thank you for your comment. Following the reviewer's suggestion, we have revised "the ground-based" as "a ground-based phenocam data" through the revised manuscript.

**[Comment 6]:** largest correlation and accuracy in comparison with what?

**Response:** The REA-based phenological dates were compared with the phenocam dates. We updated the statement as "*The start of growing season and the end of growing season in the newly merged dataset had the largest correlation (0.84 and 0.71, respectively with phenocam data)*", please refer to line 17-18 in the revised manuscript.

**[Comment 7]:** root mean square error between what? (Observed and predicted dates?)

**Response:** We have completed the statement here as "*accuracy in terms of the root mean square error (12 and 17 d, respectively between phenocam data and merged datasets)*". Please refer to line 18-19 in the revised manuscript.

**[Comment 8]:** When giving the start and end of the growing season trends, what region are you referring to – the entire globe?

**Response:** Our study focuses on the region above 30° North latitude. In the revised manuscript, we clarify the region and modified the statement as "*The new dataset has a spatial resolution of 0.05° and covers the period from 1982 to 2020, with geographic coverage extending above 30 degrees North in the Northern Hemisphere.*". Please refer to line 14-15 in the revised manuscript.

**[Comment 8]:** They still are used – I suggest changing to "are commonly".

**Response:** the terms were updated following the reviewer's suggestion, please refer to line 31.

**[Comment 9]:** What region was assessed to see trends in SOS and EOS in these studies? The entire globe?

**Response:** Thank you for your comment. The study region is in the northern hemisphere. Please refer to line 42-45 in the revised manuscript.

**[Comment 10]:** "merits and demerits" is awkward. Perhaps replace with "advantages and disadvantages"

**Response:** Following the reviewer's suggestion, we have replaced "merits and demerits" with "advantages and disadvantages" in the revised manuscript, please refer to line 45-46.

**[Comment 11]:** Perhaps set up/explain the merged dataset a little more here. From this sentence, my initial thought was "Why would a merged dataset be better if the raw datasets that go into it are inaccurate?" I understand it better after the next paragraph when you explain REA, but it's unclear here.

**Response:** We appreciate the reviewer's suggestions for improving our text. We have modified our text as "Because it is difficult to determine the optimal dataset from the various phenology datasets, producing a merged dataset using method which can choose the best dataset in different time and space among all input datasets is therefore essential for providing a comprehensive and accurate estimation of vegetation phenology with high spatiotemporal resolution." Please refer to line 51-54 in the revised manuscript.

**[Comment 12]:** change "was" to "is"
**Response:** Following the reviewer's suggestion, we have change "was" to "is", please refer to line 55.

**[Comment 13]:** What is the "vegetation index method"? Please explain in the text.
**Response:** Sorry for the confusing argument. The vegetation index method refers to the utilization of indices such as the Normalized Difference Vegetation Index (NDVI) and the Enhanced Vegetation Index (EVI) to assess and extract the status of vegetation. We have modified our text as "*Alternatively, methods such as weighted functions, the Bayesian approach, and mixed models have been combined with the vegetation index method, which used the mathematical formulas to assess vegetation conditions to integrate datasets with high temporal and spatial resolutions*". Please refer to line 58-60 in the revised manuscript.

**[Comment 14]:** Please write out all satellite/dataset abbreviations (MODIS, VIP, GIM) the first time (e.g., MODIS = Moderate Resolution Imaging Spectroradiometer)
**Response:** Thank you for your comment. Following the reviewer's suggestion, the full names of abbreviations were provided when it was first used, and the abbreviation was used when it was mentioned again in the following text. Please refer to line 76-85 in the revised manuscript.

**[Comment 15]:** I don't think you need the "Note" under the table. The dataset names and abbreviations are clear in the table.
**Response:** We removed the 'Note' section following the reviewer's suggestion.

**[Comment 16]:** Write out and define NDVI the first time it's used.
**Response:** Following the reviewer's suggestion, we have added the full name of NDVI the first time it is used. Please refer to line 50 in the revised manuscript.

**[Comment 17]:** What is the "threshold method"? You use this term multiple times, so it would be good to explain it in more detail the first time.
**Response:** We have added an explanation of the threshold method the first time it is mentioned in the text. The threshold method defines the growing state of the vegetation as the time when the vegetation index reaches a certain percentage of the annual amplitude and reflect a specific vegetation physiological growth stage. Please refer to line 91-93 in the revised manuscript.

**[Comment 18]:** What does "segment EVI2 amplitude" mean? When the time series reaches 15% of the maximum seasonal amplitude? Please add more detail about these methods.

**Response:** The definition of greenup and dormancy is officially given by the USGS (https://lpdaac.usgs.gov/products/mcd12q2v061/). "Segment EVI2 amplitude" refers to the range of variation in the Enhanced Vegetation Index 2 (EVI2) over a specific growing season segment. It is calculated as the difference between the maximum and minimum values of the EVI2 time series within that segment. To clarify this statement, in the revised manuscript, we added detail information as "*This amplitude is calculated as the difference between the maximum and minimum values of the EVI2 time series within the growing season.*". Please refer to line 94-97 in the revised manuscript.

**[Comment 19]**: What threshold was used to determine SOS and EOS? If the datasets use different thresholds, this could be one reason they differ in their SOS and EOS dates.
**Response:** Thank you for your comment. The start (end) of season is defined as the date when the NDVI2 time series first (last) crosses 35% of the segment NDVI2 amplitude. The 35% threshold using for the NDVI2 was found to be more accurate especially in regions with a protracted slow emerging growing season (see Didan, K., Barreto, A., 2016). Please refer to line 103-106 in the revised manuscript. Indeed, different methods with different thresholds result large variation in phenological dates, and thus a reliable integrated method is needed, please see our response to the similar comment#2 for detailed information.

**[Comment 20]**: This dataset uses a different threshold (20%) than the MCD12Q2 dataset (15%). As I mentioned above, this could contribute to their differences in accuracy. Also, please explain what "segment NDVI amplitude" means.
**Response:** Please refer to our response to the similar comment#2 and comment#19.

**[Comment 21]**: Why is SOS 20% and EOS 50%? Usually, the same threshold is used for both the start and end of season
**Response:** Different algorithm of phenology extraction can cause differences in vegetation phenology results. In their study (Chen and Fu, 2024), SOS 20% and EOS 50% were used as the best reflection of the state of surface vegetation.

Chen, S., Fu, Y. H., Li, M., Jia, Z., Cui, Y., and Tang, J.: A new temperature–photoperiod coupled phenology module in LPJ GUESS model v4. 1: optimizing estimation of terrestrial carbon and water processes [data set], *Geoscientific Model Development*, 17, 2509–2523, https://doi.org/10.5194/gmd-17-2509-2024, 2024.

The dataset for this study: Chen, S. and Fu, Y.: Vegetation phenology data based on GIMMS4g NDVI from 1982 to 2020, https://doi.org/10.5281/zenodo.11136967, 2024.

**[Comment 22]**: PhenoCam with capital C is used to indicate data from the PhenoCam Network (phenocam.nau.edu) and phenocam (lowercase p and c) indicates a generic camera from any network, which I would suggest using here to avoid confusion. Please see Richardson 2023, Box 3 for further explanation
(https://www.sciencedirect.com/science/article/abs/pii/S0168192323004410).
**Response:** Following the reviewer's suggestion, we used 'phenocam' instead of 'PhenoCam' through the revised manuscript.

**[Comment 23]**: Please add source of data (PhenoCam Network) "includes data

acquired from the \*PhenoCam Network\*···."

**Response:** The sources of data were provided in the revised manuscript, please refer to line 127-132 in the revised manuscript.

**[Comment 24]**: The PhenoCam Network does not use fisheye cameras – they use standard camera lens. They also aren't completely downward-facing- they are tilted slightly downwards, but always include the horizon. See Fig 5: https://phenocam.nau.edu/pdf/PhenoCam_Install_Instructions.pdf

**Response:** Thank you for your correction. We have corrected our text as "*the PhenoCam Dataset v2.0 (Richardson et al., 2018b; Seyednasrollah et al., 2019a, b), includes data derived from conventional visible-wavelength automated digital camera imagery through PhenoCam Network (Richardson et al., 2018a) over the period 2000–2018 and across 393 sites in various ecosystems, for detailed information, please refer to https://daac.ornl.gov/VEGETATION/guides/PhenoCam_V2.html and https://phenocam.nau.edu/webcam/* " please refer to line 127-132 in the revised manuscript.

**[Comment 25]**: include full link to dataset

(https://daac.ornl.gov/VEGETATION/guides/PhenoCam_V2.html). Please also cite PhenoCam data paper associated with this dataset:

> Seyednasrollah, B., A.M. Young, K. Hufkens, T. Milliman, M.A. Friedl, S. Frolking, and A.D. Richardson. 2019. Tracking vegetation phenology across diverse biomes using PhenoCam imagery: The PhenoCam Dataset v2.0. Manuscript submitted to Scientific Data. https://www.nature.com/articles/s41597-019-0229-9

**Response:** Following the reviewer's suggestion, we provided the linkage and cited the paper in the revised manuscript, please refer to line 127-132 in the revised manuscript.

**[Comment 26]**: What geographic area is included in these datasets?

**Response:** These datasets are located in Japan, and the inclusion of this portion of data is intended to supplement the validation data for Asia. Please refer to line 137 and 141 in the revised manuscript.

**[Comment 27]**: Is this dataset also collected by digital cameras?

**Response:** Yes, it is collected by digital cameras.

**[Comment 28]**: How were these 280 sites selected (for example, the PhenoCam Network has 393 sites alone)? How many sites are from each of the 3 networks?

**Response:** We use phenocam data with roi_id (a numeric code to distinguish between multiple ROIs of the same vegetation type at a given site) equals to 1000, and delete sites which only have one direction of transition dates, 26 sites was deleted in this step. And then we remove sites with no phenology values in all four data sources (90 sites), therefore 277 sites left remain out of 393, then we added 3 sites in Japan to supplement Asian area. In the revised manuscript, we clarify the selection procedure, please refer to line 145-147.

**[Comment 29]**: How are SOS and EOS extracted from all the phenocam datasets? In general, the methods of data collection and extraction from all 3 phenocam datasets could be explained in more detail.

**Response:** Following the reviewer's suggestion, in the revised manuscript, we provided

more details of phenological dates extraction methods as: *"Spline interpolation method was applied to PhenoCam data to extract transition dates for each ROI mask in PhenoCam Dataset v2.0. We used the date when the GCC first (last) crosses 25% of the GCC amplitude as the SOS and EOS. The second phenocam dataset is from the Japan Internet Nature Information System digital camera data (http://www.sizenken.biodic.go.jp/) acquired over the period 2002 – 2009. Ide and Oguma (Ide and Oguma, 2010) provided greenup dates for two phenocam sites with areas of interest (AOI) defined at the species level scale. The vegetation types included in their data comprise wetland and mixed deciduous forest. The date of green-up each year was estimated as the DOY of the maximum rate of increasing 2G-RBi (i.e., the maximum of the second derivative. The third dataset consists of phenology data for deciduous broadleaf forests in Japan (Inoue et al., 2014), supported by the Phenological Eyes Network (http://www.pheno-eye.org/), which is a network of ground-based observatories for long-term automatic observation of vegetation dynamics established in 2003 (Nasahara and Nagai, 2015), the start and end of season is defined as the first day when 20% of leaves had flushed and the first day when 80% of leaves had fallen in the given ROI, respectively."*(line 134-145).

**[Comment 30]**: How was interannual variability used to assign weights to each dataset? Are datasets considered more or less accurate with higher interannual variability? Why?

**Response:** Thank you for your comment. Interannual variability is measured by $\varepsilon_{Phe}$ in equation (2), which is also represents for natural variability. Natural variability changes from region to region, in Equation (1) and (6), $\varepsilon_{Phe}$ cancels out under the condition of $B_{Phe,i}$ and $D_{Phe,i}$ greater than $\varepsilon_{Phe}$,which based on the assumption that more stringent on are required to increase the reliability over regions characterized by lower natural variability. The natural variability does not work single, it works with $B_{Phe,i}$ and $D_{Phe,i}$ jointly. For the region in lower natural variability, if the phenology data from one dataset also have large difference with other datasets, it is given lower weight for generate the REA phenology at that region, which is thought to be less accurate data at that region. In the revised manuscript, we updated the description to clarify this issue, please refer to line 189-194 and 201-204.

Giorgi, F. and Mearns, L. O.: Calculation of Average, Uncertainty Range, and Reliability of Regional Climate Changes from AOGCM Simulations via the "Reliability Ensemble Averaging" (REA) Method, *Journal of Climate*, 15, 1141–1158, https://doi.org/10.1175/1520-0442(2002)015<1141:COAURA>2.0.CO;2, 2002.

Lu, J., Wang, G., Chen, T., Li, S., Hagan, D. F. T., Kattel, G., Peng, J., Jiang, T., and Su, B.: A harmonized global land evaporation dataset from model-based products covering 1980–2017, *Earth System Science Data*, 13, 5879–5898, https://doi.org/10.5194/essd-13-5879-2021, 2021.

**[Comment 31]**: How exactly were the datasets compared to get a value of "consistency" and "offset"? Please explain these methods in more detail.

**Response:** The consistency is measured by $B_{Phe,i}$ ,which is defined by is defined as the difference between the input dataset and the mean value of the four datasets. The offset is measured by the $D_{Phe,i}$, which is measured by the difference between the REA result and each input dataset. They are calculated in iterations. We used consistency and offset to summarize the process of $B_{Phe,i}$ and $D_{Phe,i}$. Following the reviewer's suggestion, we

revised this description, please refer to line 158-160 in the revised manuscript.

**[Comment 32]**: How are the methods described in section 2.2 and 2.2.1 different? Isn't 2.1 describing the REA method? (Also missing a 2.2.2 section in the paper).
**Response:** We are sorry for missing serial number. We have moved the contents of part 2.2 to 2.2.1 and correct the serial number in the following section.

**[Comment 33]**: What is the "voting principle"?
**Response:** The result of REA is determined by the interannual variability of each phenology dataset, together with the degree of consistency and offset among the four phenology datasets. Voting principle means that the REA result in one region is depend on all four datasets. Each dataset is given different weight to generate the result, which is determined by the differences among four datasets and its natural variability, and the result will closer to where most of data is. To clarify this issue, we revised the text, please refer to line 166-169 in the revised manuscript.

**[Comment 34]**: What does BIAS stand for? What is it measuring? Similarly, what is the difference between the RMSE and unbiased RMSE (For both, I don't mean mathematically, but rather contextually in terms of understanding the data).
**Response:** RMSE is calculated as the square root of the average of the squares of the residuals, which penalizes larger errors than smaller ones and provide an estimate of the magnitude of errors between remote sensing estimated value and phenocam datasets. BIAS is the average difference between remote sensing estimated value and phenocam value, that helps in understanding whether the estimated value is higher or lower than phenocam value. The correlation coefficient measures the linear relationship between two variables. The ubRMSE measures the deviation between two variables without systematic errors. Standard deviation quantifies the variation of the dataset, which measures the deviation between data and the mean value. Please refer to line 217-223 in the revised manuscript.

**[Comment 35]**: Were all analyses only for above 30 degrees N? If so, that should be stated in the methods.
**Response:** Thank you for your suggestion, we have stated the spatial range of the dataset in method section, e.g. 2.1 Phenology dataset part. Please refer to line 84-85 in the revised manuscript.

**[Comment 36]**: Remove the "etc" (not clear what it refers to – just list more places if desired).
**Response:** We have removed the "etc".

**[Comment 37]**: Include references for this claim.
**Response:** Following the reviewer's suggestion, the relevant references were added, please refer to line 304-306 in the revised manuscript.

**[Comment 38]**: Which site did you chose? From which phenocam dataset? Change "American" to "US".
**Response:** We chose Morganmonroe site from phenocam in figure 6, and in the revised manuscript we moved it to Fig. S4 since it is just a one site example. Following the

reviewer's suggestion, we have changed "American" to "US.".

**[Comment 39]**: Remind readers that an "advance" in SOS means earlier dates. The terms "advance" and "delay" can sometimes be confusing to follow.
**Response:** Thank you for your suggestion, we have added explanation in 3.4 to remind readers "advance" means earlier dates, and "delay" means later dates in the revised manuscript, please refer to line 356 and 360.

**[Comment 40]**: Some regions also show a significant delay in SOS (Fig 7b) – it would be good to point that out, too.
**Response:** Following the reviewer's suggestion, a description of the proportion of significant delays has been added in Fig.6 and corresponding text in the revised manuscript, please refer to line 339-341.

**[Comment 41]**: Line 329: change "were" to "are"- they're still widely used.
**Response:** We have changed our text to correct the Syntax.

**[Comment 42]**: What does the "complexity of surface backgrounds" mean?
**Response:** Complexity of surface background refers to the intricacy and variability of the land cover type in the surface, which would bring challenges in the process of extracting physiological characteristics of vegetation.

**[Comment 43]**: Please explain what the "mixed-pixel effect" is.
**Response:** The mixed-pixel effect refers to the phenomenon where a single pixel in a remote sensing image encompasses various vegetation types. This can lead to significant discrepancies in phenological dates when comparing datasets with lower spatial resolution to those with higher resolution, as the latter can capture finer details and more accurately represent the true vegetation conditions. To clarify this issue, we revised the text in the revised manuscript, please refer to line 391-394.

**[Comment 44]**: What does the "complexity of surface backgrounds" mean?
**Response:** Please see the response to comment#42 for the same question.

**[Comment 45]**: What does the "process of coefficient determination" mean?
**Response:** It refers that algorithm performance are related with their coefficients which may differs in different place, and the process of coefficient may influence the results. We revised the text to clarify this issue, please refer to line 406-409 in the revised manuscript.

**[Comment 46]**: Please further explain what the non-linear change in endmembers is and why that would result in poor performance. This is the first time it is mentioned in the paper.
**Response:** The nonlinear mixing effect of endmembers refers to the phenomenon in remote sensing imagery where the spectral signals of different land covers mix spatially in a nonlinear manner, causing the spectral response of a single pixel to no longer be a simple linear combination of the endmember spectra. The non-linear mixing of spectra caused the changes in remote sensing data reflectance, and then increase the uncertainty of vegetation indexes calculation. To clarify this issue, we revised the corresponding

text, please refer to line 410-413 in the revised manuscript.

**[Comment 47]**: Please redefine REA acronym the first time it's used in the discussion section.
**Response:** We have redefined the REA acronym, please refer to line 400 in the revised manuscript.

**[Comment 48]**: What is the "voting principle"?
**Response:** Please see the response to comment#33 for the same question.

**[Comment 49]**: It's not clear what "high processing efficiency" means – how does that relate to the low RMSE of the REA dataset?
**Response:** It means that the REA method has higher processing efficiency comparing with common data fusion method. We have removed "high processing efficiency", since it is not related to the low RMSE of the REA dataset for the revised manuscript.

**[Comment 50]**: How does this rate compare to other studies? Include citations.
**Response:** Please see the response to comment#4 for the same question.

**[Comment 51]**: I suggest replacing "invaluable" with "accurate" or "reliable" (or something similar)
**Response:** Following the reviewer's suggestion, we have changed the "invaluable" to "reliable".

**[Comment 52]**: Consider starting a new sentence here.
**Response:** Following the reviewer's suggestion, we changed the text as "*The mean uncertainty range of merged SOS and EOS dates, calculated using Equation (6), is presented in Figure 4. This range was determined using the REA method over the period from 1982 to 2020.*" Please refer to line 307 in the revised manuscript.

**Figures**
**[Comment 53]** (Fig 2): Why are plots b & d sharing an axis with plots a & c? They don't appear to share axis values. I was confused at first thinking that the weights (a & c) were shown by latitude, but that doesn't seem to be the case, so it is misleading to share an axis.
Also, the vertical figure legend is hard to read- is it possible to place it somewhere where the dataset names can be written horizontally?
**Response:** It may lead to misunderstanding with the border lines joined together. We separate a/c with b/d by discontinuous the line. And due to the limited space, placing them horizontally might require a smaller font, and we switch the way we placed the legend before in Figure 2.

[Figure]

**Figure 2: (a and c) Weights of the four phenology datasets during 1982–2020 and (b and d) latitudinal differences for (a and b) the SOS and (c and d) the EOS.** The four datasets comprise the GIM_4g, MCD12Q2, VIP, and GIM_3g datasets (for the full names, see Table 1).

**[Comment 54]** (Fig 3): Please add a label/title to the legend scale bar (something like "Proportion of contribution")

**Response:** Following the reviewer's suggestion, we added "Rate of Contribution" to the legend scale bar, please see the updated figure3.

**[Comment 55]** (Fig 4): The legends and text are small and hard to read. In panel a, consider using a color scale with more variation – it's hard to see differences between the shades of green. In the figure legend text, "EOS dates" panel should be "d" instead of "b".

**Response:** Following the reviewer's suggestion, we have changed the color of Fig 4, and revised the legend text.

[Figure]

**Figure 4: Merged mean (a) SOS and (d) EOS dates (DOY) obtained using the REA method for the period 1982–2020 and the uncertainty in the REA merged data.** Mean uncertainty ($\delta_{Phe}$) of SOS dates (b) and EOS (e) obtained using the REA method for the period 1982–2020, and its coefficient of variation (CV) in merged SOS (c) and EOS dates (f).

**[Comment 56]** (Fig 5): For all panels, include units in axes labels (DOY). In figure legend, remind readers that each data point represents a site year. I suggest moving the radar charts (panels f & k) to a separate figure (they're too small and hard to read) and include an explanation about how to interpret them in the results section.

**Response:** Following the reviewer's suggestion, we have added the "DOY" in axes labels, and Fig.5 has been redrawn for clarity, please see the updated Fig.5.

[Figure]

**Figure 5: Scatterplots and radar charts of performance for each phenology dataset and the merged phenology dataset obtained using the REA method. (**a–f) SOS evaluation results of the GIM_4g, MCD12Q2, VIP, GIM_3g, Average, and REA datasets, respectively, (m) radar chart of the SOS evaluation results, (g–l) EOS evaluation results of the GIM_4g, MCD12Q2, VIP, GIM_3g, Average, and REA datasets, respectively, and (n) radar chart of the EOS evaluation results. Each point represents a site year in the figure. OBS indicates ground-based phenocam phenological dates, RMSE indicates the root mean square error, UbRMSE indicates the unbiased RMSE, BIAS indicates the mean difference between the satellite-based results and the ground-based verification results, STD indicates the standard deviation, and CORR indicates the correlation coefficient.

**[Comment 57]** (Fig 6): Just an observation - for EOS, REA estimates are consistently late compared to phenocam. It could be related to how the EOS date is determined (which method/threshold used) for phenocam vs REA.

**Response:** It is true that if the time series data is reconstructed in the same method, the larger the threshold used may result in later SOS and earlier EOS. But datasets we used here are not reconstruct in the same method (see the revised data introduction section for details). The result of REA method is most similar to that of PhenoCam. And the PhenoCam EOS are not always lower than other datasets in all sites. Since it is difficult to observe autumn phenology on a large scale, there may be differences between satellite observation results and phenocam, the phenocam uses high frequency digital camera images to monitor vegetation phenology and is able to capture subtle changes in phenology, and remote sensing data is acquired less frequently and may not capture the exact date of EOS. We change another site and move it to supplementary materials to avoid confusion

[Figure]

**Figure S4 :Time series and box plots of Morganmonroe site data with each phenology dataset and the merged phenology dataset obtained using the REA method. (**a-b) SOS time series and box plot of the PhenoCam, GIM_4g, MCD12Q2, VIP, GIM_3g, and REA datasets, respectively, (c-d) EOS time series and box plot of the PhenoCam, GIM_4g, MCD12Q2, VIP, GIM_3g, and REA datasets, respectively.

**[Comment 58]** (Fig 7): Need to add x-axis label. Are the black lines the average SOS/EOS date for each year? Please note in figure legend what the black and red dotted lines represent. In the percentage insets in panels b and d, I assume the x-axis letters represent the significant/non-significant advance and delays (abbreviations aren't defined)? Perhaps include the abbreviations in the legend with the colors: e.g., significant delay (DS), non-significant delay (DN), etc.

**Response:** We thank the reviewer for the detailed suggestions, and following the reviewer's suggestion, we have updated the fig 7 (now fig 6). In details, red lines in (a) and (c) are the fitting lines of average SOS/EOS date for each year, blue lines in (a) and (c) are the fitting lines of average SOS/EOS dates during 1982-2020, and black lines are the average SOS/EOS date for each year. In Figure6 (a) and (c) share the x label in the middle, we have added x-axis label to make the figure clear, and added the abbreviations for figure7 (c) and (d) in the figure illustration.

[Figure]

**Figure 6: Temporal and spatial trends of the SOS and the EOS over the period 1982–2020 based on the merged dataset obtained using the REA method.** (a) Temporal trend of the SOS over the period 1982-2020, (b) Spatial trend of the SOS over the period 1982-2020, (c)Temporal trend of the EOS over the period 1982-2020, (d) Spatial trend of the EOS over the period 1982-2020. The shaded area in (a) and (c) indicates uncertainty at one standard deviation, red lines in (a) and (c) are the fitting lines of average SOS/EOS dates for each year, and black lines are the average SOS/EOS date for each year. Significant delay (DS), non-significant delay (DN), significant advance (AS), non-significant advance (AN).

---

## Author Response (AR3)

**Response to topic editor:**

**[Comment 1]** Thank you for your resubmission and patience in the rereview process. While both reviewers appreciated the authors' revisions and have a more positive outlook on the study, they still have remaining comments that need to be addressed before we consider the manuscript for publication.

**Response:** We sincerely appreciate the editor's time and effort in handling our manuscript and thank the reviewers for their thoughtful reassessment of our work, we have now included our appreciation in the acknowledgments section of the revised manuscript. We are grateful for their constructive feedback and are pleased to hear that they have a more positive outlook on our study. We have carefully revised the manuscript accordingly. Below, we provide a detailed response to each comment, outlining the corresponding revisions.

**[Comment 2]** Both reviewers still have reservations about combining the phenologic information across four datasets that use inherently different approaches. Both have specific thoughts on these points and I agree all should be addressed. At the very least, there needs to be some more discussion that combining these datasets across different methodologies is a limitation, even despite some benefit from the REA method. I noticed one or two sentences on this in the discussion, but given the concerns of the reviewer and myself, it could be expanded on a bit more. Specifically, there are both differences in the data sources, but also the methodology used from these sources. I, however, can appreciate that there are different approaches to estimating start and end of season, with not much agreement or motivation across the community about which method is "best." Integrating across methods is valuable. Indeed, also showing how REA has higher value than arithmetic average is useful and shows some potential mitigation of these issues.

**Response:** We sincerely appreciate the editor's and reviewers' insightful comments and concerns regarding the integration of phenological information across datasets that employ inherently different approaches.

We have expanded the discussion section to provide more examination of these limitations by explaining the differences in data sources and processing methodologies among the four phenology datasets, clarifying how these variations may influence the REA results. While discrepancies may exist in multi-source data integration, the REA method mitigates these inconsistencies by dynamically weighting datasets based on their interannual variability and agreement with others.

We conducted sensitivity analyses using different dataset combinations and time spans to evaluate the robustness of this method. Our results (see Figures S6 and S7) indicate that the composition of input datasets has a greater influence on fusion results than the length of the time series. The average deviation between different dataset combinations and the long-term four-dataset fusion results is 7.1 days, whereas SOS and EOS variations across different time spans (e.g., 5-year vs. 10-year fusion) are relatively minor (4.1 and 3.2 days, see Figure S6). Moreover, fusion weights derived from shorter time series are largely consistent with those obtained using the full dataset (see Figure S8). In the multi-time span analysis, the fusion results of SOS are more stable than EOS, and the longer time series result is more similar to the REA

fusion result (1982-2022).

[Figure]

**Figure S6: Comparison of fusion results of four groups of data at different time lengths from 2001 to 2010.** (a, c) Difference plots between the fusion results of SOS and EOS for the four datasets over the 2001-2010 period and the original long-term REA fusion results. (b, d) Difference plots between the fusion results of SOS and EOS obtained by separately fusing the two sub-periods of data (2001-2005 and 2006-2010) and subsequently concatenating them, and the original long-term REA fusion results.

[Figure]

**Figure S7: Differences between the SOS and EOS data fusion results from various data source combinations and the fusion results of the long-term four datasets over the 2001-2010 period.** (a- b) Group 1 represents the difference between the fusion of GIM_4g and VIP data and four datasets

REA result. (c-d) Group 2 represents the difference between the fusion of VIP data and GIM_3g data and four datasets REA result. (e-f) Group 3 represents the difference between the fusion of GIM_3g and MCD12Q2 data and four datasets REA result.

[Figure]

**Figure S8: Comparison of fusion data weights for the four datasets over different time lengths**

**from 2001 to 2010.** (a, c) Weights of the SOS and EOS fusion data for the four datasets over the entire

2001-2010 period. (b, d) Weights of the fusion data after separately fusing the two sub-periods of data (2001-2005 and 2006-2010) and subsequently concatenating them. The red dashed line separates the two sub-periods.

Our analysis suggests that datasets incorporating more reliable sources could improve the result of data fusion. For example, groups 2 and 3, which include GIM_3g SOS, exhibit smaller differences from the long-term fusion results than group 1 and demonstrate higher consistency with PhenoCam observations (Figure 5d). The EOS fusion results are more stable across different dataset combinations. These results show that multi-source data integration could help improve accuracy. The similarity in weight distributions across different dataset combinations (Figure S9) suggests that dataset selection remains a critical factor influencing final results, and incorporating additional high-quality datasets in the future could further enhance accuracy and robustness.

The effectiveness of data fusion results depends on the complementarity and quality of input datasets. The REA method can effectively integrate different datasets with independent information, enhancing both the accuracy and consistency of the phenological result. Given the lack of consensus on the "best" approach for extracting phenological dates, our study highlights the value of integrating multiple-method datasets rather than relying on a single dataset.

Furthermore, we clarify the advantage of the REA method over a simple arithmetic average by demonstrating its ability to refine SOS and EOS estimates through dynamic weighting. Our results show that REA reduces biases associated with individual datasets, producing a more reliable phenology dataset. We have made these points clearer in the revised manuscript.

[Figure]

**Figure 5: Scatterplots and radar charts of performance for each phenology dataset and the merged phenology dataset obtained using the REA method. (**a–f) SOS evaluation results of the GIM_4g, MCD12Q2, VIP, GIM_3g, Average, and REA datasets, respectively, (m) radar chart of the SOS evaluation results, (g–l) EOS evaluation results of the GIM_4g, MCD12Q2, VIP, GIM_3g, Average, and REA datasets, respectively, and (n) radar chart of the EOS evaluation results. Each point represents a site year in the figure. OBS indicates ground-based phenocam phenological dates, RMSE indicates the root mean square error, UbRMSE indicates the unbiased RMSE, BIAS indicates the mean difference between the satellite-based results and the ground-based verification results, STD indicates the standard deviation, and CORR indicates the correlation coefficient.

[Figure]

**Figure S9: Distribution of fusion weights for SOS and EOS data from different data source combinations over the 2001-2010 period.** (a, b) Group 1: Fusion weights for the combination of GIM_4g and VIP data. (c, d) Group 2: Fusion weights for the combination of VIP data and GIM_3g data. (e, f) Group 3: Fusion weights for the combination of GIM_3g and MCD12Q2 data.

**[Comment 3]** Both reviewers also agree revisions are required in the discussion. Please follow their points, which is not limited to more discussion about the limitations of the REA method itself and limitations combining the datasets that use different methodologies.

**Response:** We have revised the manuscript following the reviewers' suggestions. Specifically, we have added a discussion on the limitations of the REA method in Section 4.2 (Strengths, Limitations, and Sensitivity Analysis of the REA Approach), expanded the discussion on dataset discrepancies and integration challenges in Section 4.1 (Integrating Multi-Source Phenology Data and Addressing Dataset Discrepancies), and provided further insights into the differences in vegetation phenology trends and key influencing factors in Section 4.3 (Consistency of SOS and EOS Trends Across Studies and Key Influencing Factors). These revisions directly address the reviewers' concerns and enhance the discussion of methodological and dataset-related limitations. Please refer to lines 419-513 in the revised manuscript and we cope the text directly, please see the details below.

[revised manuscript text omitted]

**Response to Reviewer #1:**

**[Comment 1]** I appreciate the authors' edits in response to my and the other reviewer/editor's comments. The dataset and method descriptions are much improved, including how SOS and EOS are extracted for each dataset. Figure 2 is much easier to interpret now. I also appreciate the author's response about the different SOS/EOS extraction thresholds used across the different datasets. I understand that they are meant to indicate similar phenological stages, so can be combined, though I still have reservations about this method. I think this needs to be stated more clearly in the intro and/or methods so readers understand this potential source of error, and why it still works to combine datasets with differing extraction thresholds. However, given the improvement in the merged REA dataset over the initial 4 datasets, I do think this is an effective method to improve accuracy in extracting phenological transition dates.

**Response:** We sincerely appreciate the reviewer's constructive feedback and recognition of the improvements made to the dataset and method descriptions, as well as the enhanced clarity of Figure 2. Regarding the concern about combining datasets with different SOS/EOS extraction thresholds, we have revised the Introduction section to explicitly address this issue, please refer to the second paragraph in the introduction revision in the response to comment 2.

**[Comment 2]** The discussion section still needs considerable work. As I mention in the line edits below, the first couple paragraphs describing the different datasets and data fusion methods fit much better in the introduction and/or methods sections. The discussion section should provide interpretation for the observed results with examples from the existing literature. As it is, there is very little interpretation of the results and what they mean from an ecological perspective. Finally, the authors describe a "greenness trends" analysis at the end of the discussion section, which is not described in the methods section and I'm unsure of the interpretation. How was this analysis done and why? I see that the authors added this in response to my previous comment to "compare entire seasonal trends in vegetation greenness." I meant that it would be helpful to compare annual within-year variability in greenness across the different datasets rather than just the SOS/EOS dates. However, I think this could be saved for a separate paper. I apologize for the confusion.

Finally, the manuscript should generally be edited for clarity. I've suggested some edits below, but they are not exhaustive.

**Response:** Following the reviewer's suggestions, we have reconstructed the discussion section and moved the discussion of data fusion methods and the explanation of REA methods for data fusion of different threshold extraction methods to the introduction section.

The trend of Growing Season Greenness (GSG) is calculated based on the change in the mean Normalized Difference Vegetation Index (NDVI) values within the growing season (between the SOS and EOS events). The trend analysis method used in Figure S2 (now Figure S3) is the Mann-Kendall test. Greenness, which reflects vegetation growth conditions, is typically represented by NDVI (Myneni, 1997). The differences in GSG among datasets highlight the necessity of an integrated approach for robust phenology results. The specific calculation method for GSG is detailed in the response to Comment 56.

Meanwhile, we have supplemented the limitation discussion of the REA method and influencing factors of differences in the phenology trend, please refer to "4.2 Strengths,

Limitations, and Sensitivity Analysis of the REA Approach" and "4.3 Consistency of SOS and
EOS Trends Across Studies and Key Influencing Factors".

**Introduction revision:** (line 58-88)

Data fusion methods generally include unmixing-based, weight-function-based, and Bayesian-
based approaches (Gevaert and García-Haro, 2015; Piao et al., 2019a). In vegetation phenology
studies, fusion methods based on raw remote sensing data, such as the Spatial and Temporal
Adaptive Reflectance Fusion Model (Gao et al., 2006) and the Enhanced Spatial and Temporal
Adaptive Reflectance Fusion Model (Zhu et al., 2010), are often influenced by vegetation types,
growth conditions, and methodological assumptions (Sisheber et al., 2022). These methods are
typically applied to specific regions, and their performance can be affected by nonlinear spectral
mixing, where the reflectance of vegetation endmembers (i.e., the pure spectral signatures of
distinct land cover types) changes nonlinearly, and the spectral response of a single pixel is no
longer a simple linear combination of the endmember spectra (Ma et al., 2015). The nonlinear
combination of the ground feature can degrade the accuracy of vegetation phenology extraction.
Unlike these approaches, the Reliability Ensemble Averaging (REA) method is not based on
the assumption of linear reflectance changes. Instead, it directly merges annual phenology
products based on their reliability. Compared to traditional data fusion methods, the REA
method shows its advantages by simplicity and efficiency (Lu et al., 2021) while explicitly
accounting for dataset reliability, in contrast to simple averaging methods that assume equal
reliability across datasets. The simple averaging method treats all datasets equally, despite their
uncertainties varying across time and space (Lu et al., 2021; Wang et al., 2019), which leads to
potential inaccuracies in the result. The REA method considers the temporal consistency of
vegetation phenology data and uses the voting principle (Giorgi and Mearns, 2002), which
provides convergence while preserving spatial differences, making it suitable for multi-source
data fusion.

Remote sensing vegetation phenology typically reflects key transition dates in the vegetation
growth cycle, such as the start and end of the growing season, using various vegetation indices
(e.g., NDVI, LAI, and SIF) (Cong et al., 2012; Piao et al., 2019b). Ideally, phenological dates
extracted from different methods (thresholds, derivatives, smoothing functions, and fitted
models) should accurately capture changes in actual physiological conditions (De Beurs and
Henebry, 2010). However, in existing phenology datasets, there is often no "best" definition of
these transition dates (White et al., 2009). The effectiveness of different extraction methods can
vary across regions and periods, and may not always perfectly reflect true vegetation conditions
(Cong et al., 2013; De Beurs and Henebry, 2010). For instance, in high-latitude areas,
meaningful observations are relatively sparse. If the smoothing method removes too mach
information, it may reduce the ability to extract phenological signals that accurately reflect
surface dynamics (Wang et al., 2015). Therefore, integrating multiple datasets based on their
reliability, rather than relying on a single dataset or using a simple averaging method, is a more
robust approach. Our study stresses the value of integrating different datasets, and phenology
results can be improved by applying the REA method, which assigns weights based on dataset
reliability. This method can reduce uncertainties and provide a more accurate representation of
phenological dynamics across different spatial and temporal scales.

**Discussion revision:** We have substantially revised the discussion section, please refer to the response to the editor's comment#3.

**Major concerns:**

**Line edits:**

**[Comment 3]** Line 16-17: "Improved substantially" compared to what? The 4 original data sets?

**Response:** We have revised it into "The evaluation using ground-based phenocam data from 280 sites indicated that the accuracy of the newly merged dataset was substantially improved compared to the four original datasets.". Please refer to lines 15-17.

**[Comment 4]** Line 18: put "with phenocam data" outside of the parentheses so readers immediately know what it's correlated with: "had the largest correlation with phenocam data"

**Response:** We have revised it into "The start of growing season and the end of growing season in the newly merged dataset had the largest correlation with phenocam data (0.84 and 0.71, respectively) and accuracy in terms of the root mean square error between phenocam data and merged datasets (12 and 17 d, respectively).". Please refer to lines 17-19.

**[Comment 5]** Line 20: Rather than saying an "advanced trend", I would put this into context, e.g., "the start of the growing season is occurring 0.24 days earlier per year." (Same for the end of the growing season result)

**Response:** We have revised it into "Using the new dataset, we found that the start of the growing season is occurring approximately 0.19 days earlier per year ($p < 0.01$), while the end of the growing season is occurring 0.18 days later per year ($p < 0.01$) over the period 1982-2020.". Please refer to lines 19-21.

**[Comment 6]** Line 25: Perhaps change "implications" to "impacts on"?

**Response:** We have revised it into "Global change has notably altered the timing of vegetation phenology (Ettinger et al., 2020; Zhang et al., 2022), leading to important impacts on the carbon and water cycles of terrestrial ecosystems.". Please refer to line 25.

**[Comment 7]** Line 28: Suggested re-wording: "there is large variation in spatiotemporal resolution"

**Response:** We have revised the sentence into "Various vegetation phenology datasets using remote sensing data have been produced, but inconsistencies and uncertainties arise when comparing these datasets with ground-based phenological observations, and there is large variation in spatiotemporal resolution.". Please refer to lines 26-28.

**[Comment 8]** Line 32: Perhaps start this sentence with "For example," and remove "as" after PhenoCam. Also, phenology cameras (e.g., PhenoCam) aren't really a ground-based measurement since the cameras are mounted above the canopies. It's usually considered a "near-surface" remote sensing method (see https://link.springer.com/chapter/10.1007/978-3-031-75027-4_20).

**Response:** We have revised the sentence into "For example, PhenoCam, a near-surface remote sensing method, has been operational for more than 20 years.". Please refer to lines 32-33.

**[Comment 9]** Line 36-37: This sentence is a little misleading because it implies that PhenoCam data isn't processed using standardized methods, but it actually is. All PhenoCam sites are processed the same way, allowing for consistency and comparability across locations and time periods.

**Response:** We have revised the sentence into "Additionally, remote sensing datasets and PhenoCam data, are processed using standardized methods that ensure consistency and comparability across different locations and periods.". Please refer to lines 36-37.

**[Comment 10]** Line 47: Start a new sentence here

**Response:** We have revised the sentence accordingly by starting a new sentence at this point. Please refer to line 47.

**[Comment 11]** Line 49-50: Briefly describe and/or provide an example of "different extraction methods". What are you extracting (I assume SOS and EOS?).

**Response:** We have revised it into "For estimates obtained using different extraction methods, such as varying algorithms or approaches for extracting SOS and EOS from the same satellite data, the discrepancies can exceed one month.". Please refer to lines 49-51.

**[Comment 12]** Line 50-51: Please explain what it means that the required NDVI phenology extraction threshold varies across biomes.

**Response:** We have revised it into "Additionally, the NDVI (Normalized Difference Vegetation Index) threshold required for phenology extraction varies across different biomes due to differences in vegetation types, growth patterns, and environmental conditions, which affect how NDVI values correspond to phenological events such as the start and end of the growing season.". Please refer to lines 51-54.

**[Comment 13]** Line 52-53: Please re-word this for clarity: "method which can choose the best dataset in different time and space among all input datasets"

**Response:** We have revised it into "Because it is difficult to determine the optimal dataset from the various phenology datasets, producing a merged dataset using a method that selects the most suitable dataset for different times and locations from all input datasets is essential for providing a comprehensive and accurate estimation of vegetation phenology with high spatiotemporal resolution.". Please refer to lines 54-57.

**[Comment 14]** Line 55: Please define the "simple averaging method", or could re-word to say something like "Calculating the mean value across different datasets…

**Response:** We have revised it into "The simple averaging method treats all datasets equally by calculating the mean value across different vegetation phenology datasets, despite their uncertainties varying across time and space". Please refer to lines 70-71.

**[Comment 15]** Line 57: I suggest replacing "whereas" with "but in reality"

**Response:** We have reconstructed this paragraph and modified the expression. Please refer to the third paragraph of the introduction section.

**[Comment 16]** Line 59-60: This is confusing. Please clarify the definition of the "vegetation index method"- please clarify. What are "the mathematical formulas"?

**Response:** We have reconstructed this paragraph and modified the expression, and have revised it into "Remote sensing vegetation phenology typically reflects key transition dates in the vegetation growth cycle, such as the start and end of the growing season, using various vegetation indices such as NDVI (Normalized Difference Vegetation Index) and EVI (Enhanced Vegetation Index) to assess vegetation conditions". Please refer to lines 75-77.

**[Comment 17]** Line 60-61: I'm not sure what this sentence means. What "homogenous surfaces"?

**Response:** We have reconstructed this paragraph and modified the expression and removed the "homogenous" The word "homogeneous surfaces" refers to regions with relatively uniform vegetation cover. Please refer to the third paragraph of the introduction section.

**[Comment 18]** Line 67: Also provide spatial extent of new dataset here.

**Response:** The spatial resolution of the new dataset is 0.05°, with a temporal scale spanning 1982-2020, and it covers regions north of 30°N latitude. Please refer to lines 90-91.

**[Comment 19]** Line 85: Change "was" to "were"

**Response:** We have revised the sentence (line 107).

**[Comment 20]** Line 86: Can delete last sentence. Table 1 is already referenced above.

**Response:** We deleted the last sentence.

**[Comment 21]** Line 92-93: The definition of the "threshold method" is unclear. (e.g., "growing state of the vegetation as the time when…"). Maybe it would be helpful to include a figure showing this method, even if it's just in the supplement. For example, see Fig 2 of this paper: https://www.nature.com/articles/sdata201828

**Response:** According to your suggestion, we have drawn a figure (Figure S1) of the threshold method to help readers better understand this phenological extraction method. Please refer to line 117.

[Figure]

**Figure S1: Diagram of vegetation phenology extraction using the threshold method.** The red
dotted line denotes the dynamic vegetation index threshold, which is calculated by multiplying the
given threshold parameter and the maximum and minimum range of vegetation index. The green dotted
line denotes the SOS (start of season) date, and the brown dotted line denotes the EOS (end of season)
date.

**[Comment 22]** Line 95: EOS abbreviation has not been defined in the paper yet.
**Response:** We have defined the abbreviation of EOS, end of growing season, please refer to
line 50.

**[Comment 23]** Line 104-105: "The filtering method based on confidence interval and
operational continuity algorithm were used" - What does this mean?
**Response:** The filtering method, which uses confidence intervals and an operational continuity
algorithm, was applied to reconstruct the time series curves. We revised this section as "which
uses confidence intervals and an operational continuity algorithm" to make it clear that this is
part of the filtering method, please refer to lines 128-129.

**[Comment 24]** Line 107: Here and elsewhere, does "segment" mean "annual" since each
"segment" is growing season? Consider using annual or growing season instead for clarity.
**Response:** The start (end) of the season is defined using the modified Half-Max method (White
et al., 2009) as the date when the NDVI2 time series first (last) crosses 35% of the growing
season NDVI2 amplitude. In response to your suggestion, we have replaced "segment" with
"growing season" to make it clear that the period refers to the growing season, please refer to
line 131.

**[Comment 25]** Line 114-115: This is a great description of amplitude- I would move this to the
first time you describe extracting SOS and EOS dates.
**Response:** We have moved the description of amplitude to the first time we describe extracting
SOS and EOS dates, please refer to lines 117-119.

**[Comment 26]** Line 130: remove "i.e."

**Response:** We have removed "i.e.", please refer to line 155.

**[Comment 27]** Line 131: Add "the" before "PhenoCam Network"

**Response:** We have added "the" before "PhenoCam Network", please refer to line 156.

**[Comment 28]** Line 133: Start a new sentence for "For detailed information…"

**Response:** We have started a new sentence for "For detailed information…", please refer to line 157.

**[Comment 29]** Line 136: Add "A" before "spline interpolation"; add "the" before "phenocam data"

**Response:** We have added "A" before spline interpolation and added "the" before "phenocam data", please refer to line 161.

**[Comment 30]** Line 137: Define "ROI" and "GCC". Add "the" before PhenoCam Dataset v2.0"

**Response:** We have revised the sentence. "A spline interpolation method was applied to

PhenoCam data to extract transition dates for each region of interest (ROI) mask using the green chromatic coordinate (GCC) in the PhenoCam Dataset v2.0." Please refer to lines 161-162.

**[Comment 31]** Line 140: Can you use "ROI" instead of "AOI" for consistency? Or are they fundamentally different?

**Response:** We have used "ROI" instead of "AOI" for consistency, please refer to line 165.

**[Comment 32]** Line 142: Derivative of what? GCC? Another greenness metric? (Also missing a closing parenthesis)

**Response:** The date of green-up each year was estimated as the day of year (DOY)

corresponding to the maximum rate of increase in the 2G-RBi index. This is calculated as the maximum of the second derivative of the GCC time series (the maximum of the second derivative of GCC). We have also corrected the missing closing parenthesis in the revised manuscript, please refer to lines 168-169.

**[Comment 33]** Line 147-148: "no phenology values in four data sources"- what does this mean?

Are the four sources the 4 remote sensing datasets? Does it mean that 90 phenocams weren't in this study's geographic range? Or is there another reason the remote sensing and phenocam data didn't overlap?

**Response:** The phrase "no phenology values in four data sources" refers to the 90 PhenoCam sites where the four remote sensing datasets (used in this study) did not provide corresponding phenology values. This discrepancy arises because these sites include cropland surfaces, which are often not covered by large-scale remote sensing phenology datasets, and sparse vegetation areas, where low-resolution remote sensing images struggle to detect phenological signals. We have revised it into "For use in this study, we excluded 26 sites that only provided one direction of transition dates (either SOS or EOS) and removed 90 sites where the four remote sensing datasets did not provide corresponding phenology values (due to factors such as cropland surfaces or sparse vegetation areas where low-resolution remote sensing struggles to detect phenological signals). We then selected PhenoCam data from 280 sites over the period 2000–2018, resulting in 1410 site–year combinations.", please refer to lines 173-177.

**[Comment 34]** Line 159: Be sure to specify up front that the weight of each dataset can shift through time.

**Response:** We thank the reviewer for their suggestion. To clarify, the weighting method was applied to derive more accurate Start of Season (SOS) and End of Season (EOS) dates from the four vegetation phenology datasets. The weight assigned to each dataset is not fixed but can vary over time, depending on the interannual variability of each dataset and the degree of consistency and offset among the four datasets. This has now been explicitly stated in the revised manuscript, "The weight assigned to each product was based on the interannual variability of each phenology dataset, as well as the degree of consistency and offset among the four datasets. Importantly, these weights can vary over time to reflect changes in dataset reliability and performance.", please refer to line 187-189.

**[Comment 35]** Line 162: what does "calculated in iterations" mean?

**Response:** The phrase "calculated in iterations" refers to the iterative process used to determine the final weight coefficients. Specifically, consistency is measured as the difference between each input dataset and the mean value of the four datasets, while the offset is measured as the difference between the REA result and each input dataset. These values are updated iteratively during the weighting process until convergence is achieved. We have revised it into "The consistency is measured as the difference between each input dataset and the mean value of the four datasets, and the offset is measured as the difference between the REA result and each input dataset. These values are iteratively calculated during the process of determining the final weight coefficients.", please refer to lines 189-191.

**[Comment 36]** Line 169-170: Add "d" to end of "generate". How can the REA method produce data that is "consistent with most of the input phenology products at the pixel level"? Isn't the idea that each of the 4 datasets are more or less accurate at different places and times? So how can the REA dataset be consistent with most/all of them?

**Response:** The REA method operates on the "voting principle," where the final result is generated by assigning different weights to the input data sources. This approach assumes that the majority of data values at each pixel are accurate, while outliers are down-weighted or excluded. As a result, the REA dataset aligns with the majority of input phenology products at the pixel level. However, it is important to note that the accuracy of the REA data varies across different locations and times. The method dynamically determines the weight of each dataset for specific places and times, ensuring that the most reliable data sources contribute more significantly to the final result. This flexibility allows the REA method to account for the varying accuracy of the input datasets across space and time. We have revised it into "The REA method, based on the 'voting principle' (where the REA result is generated by assigning different weights to the input data sources), produces data that aligns with the majority of input phenology products at the pixel level. This approach assumes that most data values at each pixel are accurate, while outliers are down-weighted or excluded.", please refer to lines 198-200.

**[Comment 37]** 196: "on" is a typo I assume- "more stringent "on" are required"

**Response:** We have revised this sentence, please refer to line 226.

**[Comment 38]** Line 204: What does "data" refer to? One dataset or data point?

**Response:** In this context, "data" refers to individual data points. If a data point shows significant discrepancies compared to others, it may indicate issues such as improper extraction methods or regional anomalies. We have revised it into "If an individual data point shows significant discrepancies compared to others, potentially caused by improper extraction methods in that region, the $B_{Phe,i}$ and $D_{Phe,i}$ will extract this variance and incorporate it, along with the natural variability $\varepsilon_{Phe}$ of the region into the weight distribution process" , please refer to line 234-236.

**[Comment 39]** Line 230: I would just write out Mann-Kendall I(instead of M-K) since it's not used much in the manuscript.

**Response:** We have revised it from M-K to Mann-Kendall, please refer to line 261.

**[Comment 40]** Line 249: Replace "lower" with "earlier"

**Response:** We have replaced it from lower to earlier, please refer to line 287.

**[Comment 41]** Line 304: This is a little confusing. Of course EOS dates are going to be later than SOS dates. By saying "Unlike the SOS data", are you trying to point out that EOS dates are more uniform in timing that SOS? Consider re-wording to make this clearer.

**Response:** We have reworded the sentence for clarity. The revised text please refer to lines 342-343."Unlike the Start of Season (SOS) dates, which exhibit greater variability, the End of Season (EOS) dates are more consistent, predominantly occurring within day of year (DOY) 270–330 (80.0%)."

**[Comment 42]** Line 312: This is unclear: "regarding the coefficient of variation (CV) in the uncertainty range of"

**Response:** We thank the reviewer for pointing out the lack of clarity in this sentence. To address this, we have revised the text as follows:

"In Fig. 4(c, f), the coefficient of variation (CV) of the uncertainty in Start of Season (SOS) and End of Season (EOS) dates from 1982 to 2020 is analyzed. More than 56% (73%) of regions exhibit a CV below 1, 31% (18%) of regions have a CV between 1 and 1.5, and only 13% (8%) of regions show a CV higher than 1.5. No evident correlation is observed between CV and latitude changes."

Additionally, we have adjusted minor punctuation and wording to improve clarity and readability, please refer to lines 351-354.

**[Comment 43]** Line 330: First time CORR is used, please define.

**Response:** We have defined CORR as correlation coefficient, please refer to line 370.

**[Comment 44]** Line 336: What does "overestimate" mean (used 2 times)- predict dates that are too late?

**Response:** In this context, "overestimate" means that the GIM_4g dataset predicts End of Season (EOS) dates that are later than the observed dates. Specifically, the GIM_4g dataset demonstrates good performance but tends to overestimate EOS, resulting in a Root Mean Square Error (RMSE) of 43 days. We have revised it to "The GIM_4g dataset shows good performance but tends to overestimate the End of Season (EOS), with predicted dates occurring later than observed, resulting in an RMSE of 43 d. Both the VIP and the GIM_3g datasets also overestimate the EOS due to their spatial and temporal distributions, with RMSEs of 46 and 35 d, respectively." Please refer to lines 375-378.

**[Comment 45]** Line 351: I suggest adding this to the end of the first sentence: "across years at a single location."

**Response:** As recommended, we have revised the sentence to: "We selected a long-term PhenoCam site (Morganmonroe) from PhenoCam to evaluate the merged dataset across years at a single location."

**[Comment 46]** Line 352-353: suggested wording: "with data from 2010-2020 for both SOS and EOS"

**Response:** As recommended, we have revised the sentence to: "We have chosen a US. PhenoCam site characterized by deciduous broad-leaved forest, with data from 2010–2020 for both SOS and EOS." Please refer to lines 393-394.

**[Comment 47]** Line 354: Largest compared to what? The original 4 datasets?

**Response:** As illustrated in the time series plot in Figure S4, the consistency between the REA results and PhenoCam data for both SOS and EOS is the highest when compared to the original four individual datasets. Please refer to lines 395-396.

**[Comment 48]** Line 365: I suggest starting a new paragraph at "Over the study area"

**Response:** We started a new paragraph at "Over the study area". (line 407)

**[Comment 49]** Lines 375- 421: This is all background information that belongs in the introduction section. For the discussion section, I suggest starting with a 3-4 sentence recap of the study's motivations/methods, and then spend the rest of the discussion interpreting the results and linking them to the literature.

**Response:** Thank you for your suggestion. We have revised the discussion section by moving the background information to the introduction, and restructuring the discussion section into four subsections 4.1 Integrating Multi-Source Phenology Data and Addressing Dataset Discrepancies, 4.2 Strengths, Limitations, and Sensitivity Analysis of the REA Approach, 4.3 Consistency of SOS and EOS Trends Across Studies and Key Influencing Factors 4.4 Conclusion and Perspective.

Following your suggestions, we have included a brief summary of the study's motivations and methodology at the beginning of the discussion. Additionally, we have expanded our discussion on the limitations of the REA method, the factors influencing differences in phenological trends across studies, and other key considerations. These revisions ensure a clearer and more structured discussion while addressing the concerns raised.

Please refer to comment 2.

**[Comment 50]** Lines 382-386: I don't think this is needed.
**Response:** We have reconstructed the discussion, please refer to comment 2.

**[Comment 51]** Line 402: What is the "final result"?
**Response:** The "final result" refers to the outcome of the Reliability Ensemble Averaging (REA)
method, which integrates multiple datasets to produce a more robust and reliable estimate. This
has now been clarified in the revised manuscript.

**[Comment 52]** Line 411: Thanks for adding this explanation, however, please specify what
"endmember" means.
**Response:** We appreciate the reviewer's comment and have clarified the term "endmember" in
the revised manuscript. Endmembers refer to the pure spectral signatures of distinct land cover
types. In this context, the reflectance of vegetation endmembers changes nonlinearly. Due to
the spatial mixing of spectral signals from different land covers, the spectral response of a single
pixel is no longer a simple linear combination of the endmember spectra. Please refer to line
63.

**[Comment 53]** Line 416: What should be available? Multiple years of data?
**Response:** There is no restriction on the minimum length of the time series, but multiple years
of data should be available to capture natural variability and maintain accuracy. We have added
a discussion of method limitation in 4.2 Strengths, Limitations, and Sensitivity Analysis of the
REA Approach. Please refer to comment 2.

**[Comment 54]** Line 430: I suggest replacing "in details" with "for example". Also, how did
you determine that SOS "significantly advanced"? Was a statistical method used? If so, please
provide p-value. If not, please re-word.
**Response:** As recommended, we have replaced "in details" with "for example" in Line 430,
and the p-value is smaller than 0.01. We have stated in the revised manuscript "Our REA-based
dataset estimates an SOS advancement of 0.19 days per year during 1982-2020 ($p<0.01$)" and
"Similarly, EOS has been delayed at a rate of 0.18 days per year ($p<0.01$)". Please refer to lines
475 and 477.

**[Comment 55]** Line 432: Why do you think there is such large variation in the rate of SOS
advance between studies. Speculate on some possible reasons (with citations).
**Response:** In Discussion 4.3 Consistency of SOS and EOS Trends Across Studies and Key
Influencing Factors, we have expanded our discussion to address the reasons for the observed
variations in the rate of Start of Season (SOS) advancement across different studies.

Global climate change has significantly influenced vegetation phenology, with a general
advancement in SOS and a delay in EOS (Piao et al., 2019a). Our REA-based dataset estimates
an SOS advancement of 0.19 days per year during 1982-2020 ($p<0.01$), aligning with previous
findings, such as $1.4 \pm 0.6$ days per decade for 1982-2011 (Wang et al., 2015) and a 5.4-day advance from 1982 to 2008 (Jeong et al., 2011). Similarly, EOS has been delayed at a rate of 0.18 days per year (p<0.01), consistent with prior estimates of 0.18 ± 0.38 days per year for 1982–2011 (Liu et al., 2016) and a 6.6-day delay from 1982 to 2008 (Jeong et al., 2011). Deviations between REA-based trends and those from individual datasets (Blunden et al., 2023) suggest that variations in data sources and study regions contribute to observed discrepancies.

Differences in SOS trends across studies arise from multiple factors, including study period, land cover changes, spatial domain, environmental heterogeneity, and methodological differences. Trend estimates are highly sensitive to the selected time window, particularly at the start and end years (Cong et al., 2013). Interannual and decadal climate variability can further influence these trends contributing to discrepancies across studies. Land cover changes, such as wildfires and deforestation (Jeong et al., 2011), can influence the extraction result of phenology, then affect SOS trends. Additionally, spatial differences in studies also influence the rate of trends in different studies, as phenology is affected by vegetation type and climate sensitivity. For instance, Jeong et al. (2011) analyzed temperate vegetation between 30°N–80°N, and Wang et al. (2015) focused on 30°N–75°N, excluding evergreen forests and managed landscapes. Since phenology responds differently across species and locations (Maignan et al., 2008), variations in study areas contribute to discrepancies. Photoperiod constraints introduce additional latitudinal differences in phenological responses to warming (Meng et al., 2021). Moreover, precipitation and other environmental factors vary spatially, further influencing SOS estimates.

Different methods of extracting phenology from remote sensing data could introduce uncertainties between different datasets (White et al., 2009; Jeong et al., 2011; Cong et al., 2013). Differences in filling missing data and filtering methods can affect the continuity of the vegetation index time series, leading to discrepancies in phenology results. Reducing these inconsistencies among different datasets and producing more harmonized data is a primary motivation for employing the REA method, which reduces uncertainties by integrating datasets based on their reliability. Recognizing these factors also explains the discrepancies in SOS/EOS trends and stresses the need for standardized methodologies in phenological trend analyses.

**[Comment 56]** Line 437- 441: I'm not sure what this analysis is. What are "seasonal trends in greenness"? This analysis needs to be described in the methods, and reported in the results. New results are generally not introduced in the discussion section. Also what are the units of the numbers reported? What is the ecological interpretation of this analysis?

**Response:** We thank the reviewer for raising this important point. The "seasonal trends in greenness" refer to the trends in vegetation greenness during the growing season, which are quantified by analyzing the changes in the mean Normalized Difference Vegetation Index (NDVI) values between the Start of Season (SOS) and End of Season (EOS) events. The trend analysis method employed here is the Mann-Kendall test, consistent with the approach used in Figure S2 (now Figure S3).

Greenness is a widely used indicator of vegetation growth, typically represented by NDVI (Myneni, 1997). The Growing Season Greenness (GSG) is calculated as the mean NDVI value within the growing season in this study, defined by the period between SOS and EOS:

$$GSG = Mean(NDVI[Date_{SOS}, Date_{EOS}]) \qquad (16)$$

$Date_{SOS}$ is the day of year value of SOS data, and $Date_{EOS}$ is the day of year value of EOS value, $Mean$ denotes calculating the mean NDVI value in the corresponding date range.

To address the reviewer's concerns, we have now included a detailed description of this analysis in the Methods section (please refer to lines 271-277). Additionally, we have clarified the units and ecological interpretation of the analysis. Specifically, the GSG values are unitless, as NDVI is a dimensionless index, and the trends in GSG reflect changes in vegetation productivity and health over the growing season.

**[Comment 57]** Line 441: Start new paragraph with "Shifts in"

**Response:** We have reconstructed the discussion section, please refer to line 500.

**Figures:**

**[Comment 58]** Figure 1: In panel a, the shades of green are hard to distinguish. Maybe consider switching to a color palette with a larger color range.

**Response:** In response to this comment, we have modified the colormap of the figure. This adjustment significantly improves the distinguishability of the shades, thereby enhancing the overall clarity and interpretability of the figure. (line 290)

[Figure]

**Figure 1: Spatial distribution of multiyear mean SOS and EOS dates from each phenology dataset:** (a-d) multiyear mean SOS dates and (e-h) multiyear mean EOS dates derived from the GIM_4g, MCD12Q2, VIP, and GIM_3g datasets, respectively.

**[Comment 59]** Figure 2 legend: For panels b and d, are the latitudinal weights averaged over the entire time period of the REA dataset? Please state this.

**Response:** We have clarified this in the figure legend by stating: "For panels b and d, the latitudinal weights are the mean values of each dataset over their respective timespans." (line 318)

**[Comment 60]** Figure 4: Consider a color scale with more variation for panel d. For panels c and f, I suggest listing the legend values in increasing order from left to right (instead of in
columns) for easier interpretation.
**Response**: Thank you for your suggestions on the modification of Figure 4. We have adjusted
the color scale in panel (d) to enhance variation for better visual distinction. Additionally, we
have modified the legend in panels (c) and (f) by listing values in increasing order from left to
right for easier interpretation. (line 355)

[Figure]

**Figure 4: Merged mean (a) SOS and (d) EOS dates (DOY) obtained using the REA method for**
**the period 1982-2020 and the uncertainty in the REA merged data.** Mean uncertainty ($\delta_{Phe}$) of
SOS dates (b) and EOS (e) obtained using the REA method for the period 1982-2020, and its
coefficient of variation (CV) in merged SOS (c) and EOS dates (f).

**[Comment 61]** Figure 5 legend: Remove "respectively". Please briefly describe what a radar
chart is.
**Response:** We have removed "respectively" in the legend. Additionally, we have briefly
described the radar chart: "The radar chart is a graphical method used to display multivariate
data in the form of a two-dimensional chart with axes starting from the same point. (line 391)

**[Comment 62]** Figure 6 legend: I recommend using "trendlines" instead of "fitting lines". For
the last sentence, specify that these abbreviations are used in the insets of panels b and d.
**Response:** We have replaced "fitting lines" with "trendlines". We also have specified that these
abbreviations are used in the insets of panels b and d. (line 414)

The abbreviations DS (significant delay), DN (non-significant delay), AS (significant advance),
and AN (non-significant advance) are used in the insets of panels b and d. (lines 415-416)

**[Comment 63]** Figure S1 legend: RMSE between these datasets and what? The phenocam dataset?

**Response:** We have stated in the legend "RMSE between the four remote sensing datasets and the PhenoCam dataset for SOS across different PFTs during 2001-2018". (line 10-14 in

Supplement material)

**[Comment 64]** Figure S2: What does the "NDVI trend" measure? In b, what does the frequency distribution show? I'm not clear on what this analysis is.

**Response:** We have revised the legend for the previous S2 (now S3) "Figure S3: Trends in growing season NDVI derived from (a) the VIP phenology dataset, (b) the GIM_4g dataset, and (c) the REA phenology dataset for the period 1982-2015. Panel (d) presents the frequency distribution of the Sen's slope for the growing season NDVI trend in regions north of 30°N. GS:

growing season. Black dots indicate regions where the trend is statistically significant (P <

0.05)." (line 20-224 in Supplement material)

Panel (d) illustrates the frequency distribution of the Sen's slope for the growing season NDVI

trend in regions north of 30°N, allowing for a comparative analysis of greenness trends across different datasets.

[Figure]

**Figure S3: Trends in growing season NDVI derived from (a) the VIP phenology dataset, (b) the**

**GIM_4g dataset, and (c) the REA phenology dataset for the period 1982 – 2015.** Panel (d) presents the frequency distribution of the Sen's slope for the growing season NDVI trend in regions north of

30° N. GS: growing season. Black dots indicate regions where the trend is statistically significant (P <

0.05).

**[Comment 65]** Figure S3 legend: Specify that the simple average is across all 4 datasets.

**Response:** We revised the legend as "Mean (a) SOS and (b) EOS dates (DOY) obtained using the simple average across all four datasets for the period 1982-2020. RMSE represents the root mean square error (days), and CORR represents the correlation coefficient. The four datasets
include MCD12Q2, VIP, GIM_4g, and GIM_3g. For further details on these datasets, please
refer to Section 2.1.". (line 26-30 in Supplement material)

**[Comment 66]** Figure S4: Specify what that dotted lines indicate (trendlines for each dataset I
assume). Specify that "morganmonroe" is a PhenoCam site and add more site details (location?
Vegetation type?). For panel a, the green dataset looks like it has a short dashed trendline for
the early years, but no data points?
**Response:** We have added "The dotted lines in a and c represent the trendlines for each dataset.
The Morganmonroe site, located in Morgan Monroe State Forest, Indiana, is characterized by
deciduous broadleaf forest." in the legend. (line 31-37 in Supplement material)

In panel (a), the green dataset's data points overlap with the REA data, which is why they may
not be clearly visible.

[Figure]

**Figure S5: Time series and box plots of Morganmonroe site data with each phenology dataset and**
**the merged phenology dataset obtained using the REA method. (**a-b) SOS time series and box plot
of the PhenoCam, GIM_4g, MCD12Q2, VIP, GIM_3g, and REA datasets, respectively, (c-d) EOS time
series and box plot of the PhenoCam, GIM_4g, MCD12Q2, VIP, GIM_3g, and REA datasets,
respectively. The dotted lines in a and c represent the trendlines for each dataset. The Morganmonroe
site, located in Morgan Monroe State Forest, Indiana, is characterized by deciduous broadleaf forest.

**[Comment 1]** Thank you for the detailed point-by-point response. I think most of my comments have been addressed.

**Response:** Thank you for your feedback. We appreciate your constructive comments and are glad that most of your concerns have been addressed.

**[Comment 2]** I checked the GitHub codes and feel that it lacks sufficient comments or guidance to ensure reproducibility.

**Response:** Following your suggestion, we have added comments in the GitHub to ensure the reproducibility of the codes.

**[Comment 3]** Regarding the response to comments 12 and 18, I recommend that the authors include a more detailed discussion of the limitations of the REA method.

**Response:** Thank you for your suggestion, we have added more details about the limitations of our method in the discussion section 4.2 Strengths, Limitations, and Sensitivity Analysis of the REA Approach.

Given the lack of consensus on the "best" approach for extracting phenological dates, our study highlights the value of integrating multiple methodologies rather than relying on a single dataset. The REA method improves phenology estimations by weighting datasets based on reliability rather than assuming equal contributions, as in a simple arithmetic average (Lu et al., 2021). The REA method considers the temporal correlation of vegetation phenology data by employing a voting principle (Giorgi and Mearns, 2002), and this approach facilitates convergence of data while retaining differences of the spatial distribution, thereby offering advantages in multisource data fusion than simple averaging. Our results indicate that REA merged phenology results are more consistent with PhenoCam observations than individual datasets and simple average results, demonstrating its potential to reduce biases between different datasets caused by different extraction methods. However, while REA enhances consistency, it does not fully eliminate methodological discrepancies.

To evaluate the robustness of this approach, we conducted sensitivity analyses using different dataset and time span combinations. Our results (see Figures S6 and S7) indicate that the combination of different input datasets has a greater influence on fusion results than the length of the time series. The average deviation between different dataset combinations and the long-term four-dataset fusion results is 7.1 days, whereas SOS and EOS variations across different time spans (e.g., 5-year vs. 10-year fusion) are relatively minor (4.1 and 3.2 days, see Figure S6). Moreover, fusion weights derived from shorter time series are largely consistent with those obtained using the full dataset, see Figure S8. In the multi-time span analysis, the fusion results of SOS are more stable than EOS, and the longer time series result is more similar to the REA fusion result (1982-2022).

[Figure]

**Figure S6: Comparison of fusion results of four groups of data at different time lengths from 2001 to 2010.** (a, c) Difference plots between the fusion results of SOS and EOS for the four datasets over the 2001-2010 period and the original long-term REA fusion results. (b, d) Difference plots between the fusion results of SOS and EOS obtained by separately fusing the two sub-periods of data (2001-2005 and 2006-2010) and subsequently concatenating them, and the original long-term REA fusion results.

[Figure]

**Figure S7: Differences between the SOS and EOS data fusion results from various data source combinations and the fusion results of the long-term four datasets over the 2001-2010 period.** (a-b) Group 1 represents the difference between the fusion of GIM_4g and VIP data and four datasets REA result. (c-d) Group 2 represents the difference between the fusion of VIP data and GIM_3g data and four datasets REA result. (e-f) Group 3 represents the difference between the fusion of GIM_3g and MCD12Q2 data and four datasets REA result.

The effectiveness of data fusion depends on the complementarity and quality of input datasets.
Our analysis suggests that datasets incorporating more reliable sources could produce improved
fusion results. For example, GIM_3g SOS demonstrate higher consistency with PhenoCam
observations (Figure 5d); groups 2 and 3, which include GIM_3g SOS, exhibit smaller
deviations from the long-term REA fusion results than group 1. In contrast, EOS estimates are
more stable across different dataset combinations. These results show that multi-source data
integration could help improve accuracy. The similarity in weight distributions across different
dataset combinations (Figure S9) suggests that dataset selection remains a critical factor
influencing final estimates. Incorporating additional high-quality datasets in future studies
could further enhance accuracy and robustness.

[Figure]

**Figure 5: Scatterplots and radar charts of performance for each phenology dataset and the**
**merged phenology dataset obtained using the REA method. (**a–f) SOS evaluation results of the
GIM_4g, MCD12Q2, VIP, GIM_3g, Average, and REA datasets, respectively, (m) radar chart of the
SOS evaluation results, (g–l) EOS evaluation results of the GIM_4g, MCD12Q2, VIP, GIM_3g,
Average, and REA datasets, respectively, and (n) radar chart of the EOS evaluation results. Each point
represents a site year in the figure. OBS indicates ground-based phenocam phenological dates, RMSE
indicates the root mean square error, UbRMSE indicates the unbiased RMSE, BIAS indicates the mean
difference between the satellite-based results and the ground-based verification results, STD indicates
the standard deviation, and CORR indicates the correlation coefficient.

[Figure]

**Figure S8: Comparison of fusion data weights for the four datasets over different time lengths from 2001 to 2010.** (a, c) Weights of the SOS and EOS fusion data for the four datasets over the entire 2001-2010 period. (b, d) Weights of the fusion data after separately fusing the two sub-periods of data (2001-2005 and 2006-2010) and subsequently concatenating them. The red dashed line separates the two sub-periods.

[Figure]

**Figure S9: Distribution of fusion weights for SOS and EOS data from different data source combinations over the 2001-2010 period.** (a, b) Group 1: Fusion weights for the combination of GIM_4g and VIP data. (c, d) Group 2: Fusion weights for the combination of VIP data and GIM_3g data. (e, f) Group 3: Fusion weights for the combination of GIM_3g and MCD12Q2 data.

**[Comment 4] Additionally, were all four datasets available all the time across 1982-2020 to be**

**merged for the new dataset? Would it be possible to assign a quality flag to the new dataset?**

**Response:** Thank you for your constructive suggestions. Based on the number of datasets involved in data fusion at different times, we assigned a quality flag (1-4, Data confidence from low to high) and uploaded it to the data website to enhance the application of the data.

---

## Author Response (AR4)

**Response to topic editor:**

**[Comment 1]** We have received an additional review from reviewer 1, and they generally find the technical aspects of REA to appear robust. However, they note several major clarity issues. The most major is that a new sensitivity analysis was added without any methodological explanation. I agree with this assessment, and agree that such an analysis needs to be fully described in the main text in the methods. They also felt that there are several locations throughout that are unclear yet, and that the messaging needs improvement. There appear to be instances where they are requesting an edit that had been requested in the previous round. Therefore, while the reviewer concerns about technical aspects appear addressed, clarity needs to be improved before the manuscript can be accepted for publication.

**Response:** We sincerely appreciate the editor's time and effort. In response to your and the reviewer's suggestions, we have added the detailed methodological explanation of the sensitivity analysis, and revised the manuscript throughout to enhance clarity and strengthen the messaging. Please find our detailed replies to each of the reviewers' comments below.

**Response to Reviewer #1:**

**[Comment 1]** Thanks to the authors for their edits and thorough response to my previous comments. The discussion of the REA method limitations and the methods for the "trend" analyses are more robust. The discussion section also has more connection to the literature now. However, it appears that a new analysis has been added without thorough explanation. In the discussion section (lines 458-473), the authors mention that they conducted a sensitivity analysis. This is the first time this analysis is mentioned in the paper. It should have been included much earlier in the methods section and the process thoroughly explained (e.g., "groups" are mentioned in lines 468-469, but there's no explanation of what a "group" is). I think I pieced together what the authors are trying to do from the figures, but there needs to be far more explanation in the text about what this is and why/how it was done. Given more context, I think this can be important piece of information to support the REA dataset credibility/robustness.

**Response:** We sincerely thank the reviewer for the positive and thoughtful suggestions, which have significantly improved the clarity and rigor of our manuscript. We appreciated the reviewer's help in the acknowledgments in the revised manuscript. Sepcifictly, we have added detailed descriptions of the sensitivity analysis earlier in the Methods section and improved the text according to the reviewer's suggests, please refer to the text in Lines 241-247, and lines 472-474, lines 476-477 in the revised manuscript.

**[Comment 2]** Also, as stated before, please also be careful with the use of "phenocam" vs "PhenoCam" throughout. Below are additional line edits.

**Response:** Following the reviewer's suggestion, we have revised the use of "phenocam" and "PhenoCam", please refer to line 33, 37, 178, 405 in the revised manuscript.

**Line edits:**

**[Comment 3]** Line 18: "largest correlation" compared to what? The original datasets?

**Response:** We have revised it as "*The start and end of the growing season in the newly merged dataset showed the highest correlation with ground-based phenocam observations, compared to the original datasets.*" (lines 17-18 in the revised manuscript).

**[Comment 4]** Line 20: Is this the average rate for the entire Northern Hemisphere? I would add that in.

**Response:** It is the average rate for the regions north of 30º N. Following the reviewer's suggestion, we have clarified this information in the revised manuscript in lines 21-22.

**[Comment 5]** Line 32-33: When you say PhenoCam here, do you mean "the PhenoCam Network" or phenocams in general? The PhenoCam Network started in 2008, so it is NOT over 20 years old yet. However, the use of phenocams more generally has occurred for over 20 years. I think you mean "phenocams" in this context.

**Response:** We are sorry for the unclear description, in the revised manuscript, we have revised this sentence as "*For example, phenocams, a near-surface remote sensing tool, has been operational for more than 20 years*" (lines 33-34).

**[Comment 6]** Line 26: change to "phenocam"

**Response:** changed following the reviewer's suggestion (line 37 in the revision).

**[Comment 7]** Line 48-49: Define acronyms AVHRR and VIPPHEN

**Response:** Defined following the reviewer's suggestion (line 49-51 in the revision).

**[Comment 8]** Line 51: This is the first time NDVI is mentioned, so please provide a brief explanation of what it measures.

**Response:** Following the reviewer's suggestion, we have provided a brief explanation of NDVI in the revised manuscript, please refer to lines 54-57.

**[Comment 9]** Line 68-69: This wording is awkward: "method shows its advantages by simplicity and efficiency

**Response:** We have revised it as "Compared to traditional data fusion methods, the REA method shows clear advantages in simplicity and computational efficiency" (lines 86-87 in the revised manuscript).

**[Comment 10]** Line 73: This is the first time the "voting principle" is mentioned. Please explain what it is here.

**Response:** Following the reviewer's suggestion, we have provided a explanation of voting principle in the revised manuscript, please refer to lines 91-92.

**[Comment 11]** Line 75-88: I suggest switching the order so that this paragraph comes before the previous paragraph. That way you describe remote sensing phenology methods before you discuss combining them using REA.

**Response:** We thank the reviewer for this thoughtful comment, and we have switched the order of these two paragraphs following the reviewer's suggestion, please refer to the lines 62-93 in the revised manuscript.

**[Comment 12]** Line 118-119: What is the "specific segment of the growing season"? Does that just mean the entire growing season? If so, please state that.

**Response:** The amplitude is calculated as the difference between the maximum and minimum values of the EVI2 time series during the entire growing season. To clarify, we updated the text in the revised manuscript, please refer to lines 122-123.

**[Comment 13]** Line 121-123: This is a repeated sentence. I suggest deleting it in this location.
**Response:** Deleted following the reviewer's suggestion.

**[Comment 14]** Line 148-149: You already defined amplitude above. I suggest deleting this sentence.
**Response:** Deleted following the reviewer's suggestion.

**[Comment 15]** Line 150: How were they produced? Were all 5 averaged? Or combined in a different way?
**Response: yes, it was averaged across the five methods. In the revised manuscript w**e revised it as: "*The average spring (SOS) and autumn (EOS) phenological dates were produced by averaging the results obtained from the five fitting methods*" (lines 150-151).

**[Comment 16]** Line 162: Briefly explain what GCC is and provide a source (e.g., a measure of greenness intensity calculated from digital imagery)
**Response:** Following the reviewer's suggestion, we updated the explanation, please refer to lines 162-165 in the revised manuscript.

**[Comment 17]** Line 165: add "the" before ROI
**Response:** Added following the reviewer's suggestion (line 167).

**[Comment 18]** Line 174: I'm confused by this- isn't the point that some remote sensing products are better for certain ecosystems than others? Isn't this biasing your comparison if you're removing these challenging sites?
**Response:** Our intention was to ensure the validity of inter-dataset comparisons by focusing on sites where all products provided usable data. Including sites with invalid phenology estimates in remote sensing data would introduce noise and potentially distort the evaluation of consistency and performance. To clarify, in the revised manuscript, we updated the sentence as follows: "*For this study, we excluded 26 sites that only provided one type of transition date (either SOS or EOS) and removed 90 sites where none of the four remote sensing datasets provided valid phenology estimates. These excluded sites were primarily located in cropland-dominated areas or regions with sparse vegetation, where the low spatial resolution of remote sensing data limits the reliable detection of phenological transitions*." (lines 174-179).

**[Comment 19]** Line 176: I think you mean "phenocam" here.
**Response:** We have revised it into "phenocam" (line 178).

**[Comment 20]** Line 260: Delete the word "hypothesis"
**Response:** Deleted following the reviewer's suggestion.

**[Comment 21]** Line 261: Please explain what the SOS/EOS "trend" is (e.g., the change in SOS/EOS date through time).

**Response:** Following the reviewer's suggestion, we updated the explaination, please refer to lines 270-273 in the revised manuscript.

**[Comment 22]** Line 271: Didn't you also use a Mann-Kendall trend test to analyze the growing season greenness trend? If so, please state that.

**Response:** yes, we also used the MK trend text. Following the reviewer's suggestion, we updated the statement in the revised manuscript please refer to lines 271-273.

**[Comment 23]** Line 337: change to: "North America and on the Qinghai–Tibet Plateau"

**Response:** Changed following the reviewer's suggestion (line 347-348).

**[Comment 24]** Line 343: end of season shouldn't be capitalized

**Response:** Changed following the reviewer's suggestion (line 353).

**[Comment 25]** Line 351: Missing parentheses around "Fig. 4(b, e)"

**Response:** Changed following the reviewer's suggestion (line 360).

**[Comment 26]** Line 353-354: Don't need to write out SOS and EOS again here.

**Response:** Deleted following the reviewer's suggestion.

**[Comment 27]** Line 369: "VIP dataset has a lower estimation in the SOS range of DOY 100–140" - what does this mean? That VIP is bad a predicting SOS dates when they occur early in the season? I guess I don't understand why the DOY 100-104 range is given here.

**Response:** Thank you for pointing this out. The VIP dataset show an earlier SOS dates compared to other datasets, particularly in regions where the actual SOS falls between day of year (DOY) 100–140, which corresponds to early spring. To clarify, in the revised manuscript, we have revised the sentence as follows: "*The VIP dataset tends to estimate earlier SOS dates than the other datasets, especially in areas where SOS occurs early in the season (DOY 100–140).*" (lines 378-379).

**[Comment 28]** Line 377: Don't need to define EOS again

**Response:** Changed following the reviewer's suggestion (line 387).

**[Comment 29]** Line 379: Better compared to what? A simple average of the datasets?

**Response:** yes. in the revised manuscript, we clarify this statements as: "The REA-based EOS also shows better performance compared to the simple average of the original datasets" (lines 389-390).

**[Comment 30]** Line 394: from "the PhenoCam Network"

**Response:** Changed following the reviewer's suggestion (line 405).

**[Comment 31]** Line 395: Remove period after US

**Response:** Changed following the reviewer's suggestion (line 405).

**[Comment 32]** Line 405: Should this say Fig 6 instead of 7?
**Response:** Thank you for pointing this mistake and it has been corrected (line 416).

**[Comment 33]** Line 428: Replace "across" with "between"
**Response:** Changed following the reviewer's suggestion (line 439).

**[Comment 34]** Line 442: Start new paragraph here - this doesn't relate to differences in ecological regions (topic sentence of this paragraph). Also, define "greening trend" (e.g., how average greenness over the growing season changes through time). Put this in context- what does it mean?
**Response:** Changed following the reviewer's suggestion (line 454-459).

**[Comment 35]** Line 458-473: See my notes above for this section. Also, line 460" "average deviation" of what? SOS and EOS?; Line 463: What is a "multi-time span analysis"?
**Response:** Changed following the reviewer's suggestion (line 241-247, 472-474, 476-477).

**[Comment 36]** Line 488-489: What was the difference in rates they found due to this difference?
**Response:** In the revised manuscript, we updated this statements as: "*For instance, Jeong et al. (2011) analyzed temperate vegetation between 30° N－80° N (SOS advance in 0.2 days per year ), and Wang et al. (2015) focused on 30° N-75° N( SOS advance in 0.14 ± 0.6 days per year), excluding evergreen forests and managed landscapes.*" (line 500-501).

**Figures**
**[Comment 37]** Figure 2 caption: It's confusing to relist the panel letters again- I suggest changing to "(a and c) Weights of the four phenology datasets during 1982－2020 and (b and d) latitudinal differences for (upper) the SOS and (lower) the EOS"
**Response:** Changed following the reviewer's suggestion (line 330).

**[Comment 38]** Figure 3 caption: Recommended re-wording "Spatial distribution of the mean contribution of the four datasets to the merged SOS and EOS results. The mean SOS (a-d) and EOS (e-h)weight derived from the GIM_4g, MCD12Q2, VIP, and GIM_3g datasets." And delete everything after.
**Response:** Changed following the reviewer's suggestion (line 344).

**[Comment 39]** Figure 4: Would it be possible to use the same color scales for panels a & d? It would make it easier to compare and interpret.
**Response:** Thank you for the suggestion, but we intentionally used different color scales to visually distinguish the spring (SOS) and autumn (EOS) phenophases, reflecting their seasonal characteristics—green tones for spring onset and brown tones for autumn senescence. We therefore prefer using the different colors.

**[Comment 40]** Figure 5 m & n: I don't see the average group on the radar charts. Is it

complexly covered by another group? Perhaps use a color that's more different from the REA group?

185 **Response:** Changed following the reviewer's suggestion (line 395).

**[Comment 41]** Figure S1 caption: I'm a little confused- what's the difference between a "given" threshold and the "dynamic vegetation threshold"?
**Response:** The "given" threshold refers to a fixed NDVI or EVI2 value, which is used to identify phenological events. The "dynamic" threshold is calculated by multiplying the given
190 threshold by the range of the vegetation index, and then adding the minimum value of the vegetation index. In the revied manscript, we have clarified the difference in the caption. (line 5)

**[Comment 42]** Figure S3 caption: Please state what a positive vs negative Sens slope means in context
195 **Response:** Changed following the reviewer's suggestion (line 20-25).

**[Comment 43]** Figure S6 caption: Include the original REA timeframe to remind readers.
**Response:** Changed following the reviewer's suggestion (line 43).

**[Comment 44]** Figure S7: How were the groups selected? It doesn't include all the possible dataset combinations.
200 **Response:** Because including all possible dataset combinations would take up too much space here, we therefore selected three representative combinations that each include all four datasets used in the complete fusion. These combinations were chosen to illustrate the influence of different data source selections on the final fusion results.